



# TROPOMI tropospheric ozone column data : Geophysical assessment and comparison to ozonesondes, GOME-2B and OMI

Daan Hubert[1], Klaus-Peter Heue[2,3], Jean-Christopher Lambert[1], Tijl Verhoelst[1], Marc Allaart[4], Steven Compernolle[1], Patrick D. Cullis[5], Angelika Dehn[6], Christian Félix[7], Bryan J. Johnson[5], Arno Keppens[1], Debra E. Kollonige[8,9], Christophe Lerot[1], Diego Loyola[2], Matakite Maata[10], Sukarni Mitro[11], Maznorizan Mohamad[12], Ankie Piters[4], Fabian Romahn[2], Henry B. Selkirk[9,13], Francisco R. da Silva[14], Ryan M. Stauffer[9,15], Anne M. Thompson[9], J. Pepijn Veefkind[4], Holger Vömel[16], Jacquelyn C. Witte[16], and Claus Zehner[6]

[1]Royal Belgian Institute for Space Aeronomy (BIRA-IASB), Ringlaan 3, 1180 Uccle (Brussels), Belgium
[2]German Aerospace Centre (DLR), Münchener Straße 20, 82234 Weßling, Germany
[3]Technische Universität München, Arcisstrasse 21, 80333 München, Germany
[4]Royal Netherlands Meteorological Institute (KNMI), Utrechtseweg 297, 3730 AE De Bilt, The Netherlands
[5]NOAA Global Monitoring Laboratory (NOAA/ESRL/GML), 1325 Broadway, Boulder 80305-3337, CO, USA
[6]European Space Agency/Centre for Earth Observation (ESA/ESRIN), Largo Galileo Galilei 1, 00044 Frascati (Roma), Italy
[7]Federal Office of Meteorology and Climatology, MeteoSwiss, Payerne, Switzerland
[8]Science Systems and Applications, Inc., Lanham, MD, USA
[9]Atmospheric Chemistry and Dynamics Lab, NASA Goddard Space Flight Center, Greenbelt, MD, USA
[10]School of Biological and Chemical Sciences, University of the South Pacific, Fiji
[11]Meteorological Service of Suriname, Paramaribo, Suriname
[12]Atmospheric Science and Cloud Seeding Division, Malaysian Meteorological Department, Petaling Jaya, Selangor, Malaysia
[13]Universities Space Research Association, Columbia, MD, USA
[14]Laboratory of Environmental and Tropical Variables, Brazilian Institute of Space Research, Natal, Brazil
[15]Earth System Science Interdisciplinary Center, University of Maryland, College Park, MD, USA
[16]National Center for Atmospheric Research, Boulder, CO, USA

**Correspondence:** daan.hubert@aeronomie.be

**Abstract.** Ozone in the troposphere affects humans and ecosystems as a pollutant and as a greenhouse gas. Observing, understanding and modelling this dual role, as well as monitoring effects of international regulations on air quality and climate change, however, challenge measurement systems to operate at opposite ends of the spatio-temporal scale ladder. On board of the ESA/EU Copernicus Sentinel-5 Precursor (S5P) satellite launched in October 2017, TROPOspheric Monitoring Instrument (TROPOMI) aspires to take the next leap forward by measuring ozone and its precursors at unprecedented horizontal resolution until at least the mid 2020s. In this work, we assess the quality of TROPOMI's first release (V01.01.05–08) of tropical tropospheric ozone column data (TrOC). Derived with the Convective Cloud Differential (CCD) method, TROPOMI daily TrOC data represent the three-day moving mean ozone column between surface and 270 hPa under clear sky conditions gridded at 0.5° latitude by 1° longitude resolution. Comparisons to almost two years of co-located SHADOZ ozonesonde and satellite data (Aura OMI and MetOp-B GOME-2) conclude to TROPOMI biases between −0.1 and +2.3 DU (<+13%) when averaged over the tropical belt. The field of the bias is essentially uniform in space (deviations <1 DU) and stable in time at





the 1.5–2.5 DU level. However, the record is still fairly short and continued monitoring will be key to clarify whether observed patterns and stability persist, alter behaviour or disappear. Biases are partially due to TROPOMI and the reference data records themselves, but they can also be linked to systematic effects of the non perfect co-locations. Random uncertainty due to co-location mismatch contributes considerably to the 2.6–4.6 DU (∼14–23%) statistical dispersion observed in the difference time series. We circumvent part of this problem by employing the triple co-location analysis technique and infer that TROPOMI single-measurement precision is better than 1.5–2.5 DU (∼8–13%), in line with uncertainty estimates reported in the data files. Hence, the TROPOMI precision is judged to be 20–25% better than for its predecessors OMI and GOME-2B, while sampling at four times better spatial resolution and almost twice better temporal resolution. Using TROPOMI tropospheric ozone columns at maximal resolution nevertheless requires consideration of correlated errors at small scales of up to 5 DU due to the inevitable interplay of satellite orbit and cloud coverage. Two particular types of sampling error are investigated and we suggest how these can be identified or remedied. Our study confirms that major known geophysical patterns and signals of the tropical tropospheric ozone field are imprinted in TROPOMI's two-year data record. These include the permanent zonal wave-one pattern, the pervasive annual and semiannual cycles, the high levels of ozone due to biomass burning around the Atlantic basin, and enhanced convective activity cycles associated with the Madden-Julian Oscillation over the Indo-Pacific warm pool. A quasi-periodic signal of 1–2 weeks and 3–5 DU amplitude in TrOC time series, especially at low latitudes, is reminiscent of Kelvin wave activity. TROPOMI's combination of higher precision and higher resolution reveal details of these patterns and the processes involved, at considerably smaller spatial and temporal scales and with more complete coverage than contemporary satellite sounders. If the accuracy of future TROPOMI data proves to remain stable with time, these hold great potential to be included in Climate Data Records, as well as serve as a travelling standard to interconnect the upcoming constellation of air quality satellites in geostationary and low Earth orbits.

## 1 Introduction

Although present only in traces (concentrations of parts per billion volume air) and representing just 10% of the total column of atmospheric ozone ($O_3$), tropospheric ozone plays a central role in the oxidation chemistry in the troposphere (Monks et al., 2015, and references therein). It is also an air pollutant: exposure to high levels of $O_3$ can cause respiratory issues and be detrimental to health, vegetation and materials. Being a greenhouse gas it is recognised as an Essential Climate Variable (ECV) for the Global Climate Observing System (GCOS, 2016), for which measurements in the long term are required on the global scale. While some ozone is released at the surface by soils and plants and some descends from the stratosphere into the upper troposphere, most of the ozone found in the troposphere is actually formed there by the interaction of solar ultraviolet radiation with hydrocarbons and nitrogen oxides, its precursors. The latter are emitted by natural processes (e.g., lightning and wildfires) and anthropogenic activities (e.g., intentional biomass burning, fuel combustion, power plants and other industrial activities). Precursors and their long-lived reservoirs can be transported over intercontinental distances to the point of ozone production. As a result, the tropospheric ozone field is highly variable over a wide range of spatial and temporal scales, which, in turn, poses a clear challenge to the observing system.





Since the late 1980s the global distribution of tropospheric $O_3$ has been inferred from satellite measurements of the ultra-violet radiance back-scattered at nadir. Tropospheric $O_3$ from space was first determined through residual techniques, which consist in subtracting from satellite total $O_3$ column data an estimate of the stratospheric component, e.g., subtracting SAGE or SBUV stratospheric columns from TOMS total columns (Fishman et al., 1990; Fishman, 1991; Hudson et al., 1995; Hudson and Thompson, 1998). Residual-type retrievals were later advanced by cloud slicing techniques, which use masking properties

of the clouds present in the $O_3$ data sets to separate the tropospheric and stratospheric components of the total $O_3$ column, or to infer ozone concentrations in a given layer of the (upper) troposphere (Ziemke et al., 1998, 2001, 2009b). The technique is particularly successful in presence of deep convective clouds and therefore the Convective Cloud Differential technique (CCD) has been a privileged approach to derive tropical tropospheric $O_3$ fields from the currently operating European satellite instruments: Aura OMI, MetOp GOME-2 and Sentinel-5 Precursor TROPOMI, and their predecessors ERS-2 GOME and

Envisat SCIAMACHY (Valks et al., 2003, 2014; Ziemke et al., 2010; Heue et al., 2016; Leventidou et al., 2016, 2018). Ensuring the continuous global monitoring of tropospheric $O_3$ beyond horizon 2040 is a requirement of the EU Earth Observation programme Copernicus (Ingmann et al., 2012). Therefore the Copernicus Space Component (CSC) plans a series of three Sentinel-5 atmospheric composition missions built by ESA and to be launched nominally in 2023, 2030 and 2037, on board of the EUMETSAT EPS/MetOp Second Generation platforms MetOp-SG-A1/2/3.

As a gap filler between heritage satellites and the Sentinel-5 series, Sentinel-5 Precursor (S5P) was launched in October 2017 with aboard the TROPOspheric Monitoring Instrument (TROPOMI, Veefkind et al. (2012)). Pre-launch mission requirements for the Sentinel tropospheric $O_3$ data target a systematic error and an uncertainty both lower than 25% (ESA, 2017a, b). Since the beginning of its nominal operation in April 2018, in-flight compliance of S5P TROPOMI tropical tropospheric $O_3$ data with pre-launch requirements has been monitored routinely by the S5P Mission Performance Centre (MPC) through comparisons

to balloon-based ozonesonde measurements and to similar CCD-derived satellite data from OMI and GOME-2B. A range of in-depth investigations has also been carried out to assess the geophysical information available in the TROPOMI data set, including analysis of uncertainty, detection of geographical and sampling patterns, triple co-location studies, and power spectrum analysis of the time series.

    The objective of this paper is to report on the outcome of this comprehensive investigation of the first two years of S5P

TROPOMI tropospheric $O_3$ column data. The TROPOMI data record under investigation, S5P L2_O3_TCL V01.01.05–08, is described in Sect. 2. Correlative measurements used as a reference in the validation studies are described in Sect. 3, as well as the applied co-location criteria. Comparisons with respect to ozonesondes and to other satellites are reported in Sect. 4. Sampling errors at small scales are determined in Sect. 5. Section 6 describes how TROPOMI captures known geophysical features like the zonal wave-one pattern, tropospheric $O_3$ enhancements associated with biomass burning, and natural cycles

like the Madden-Julian Oscillation (MJO). In Sect. 7 all results are assembled and discussed to derive conclusions on the bias, the uncertainty, the stability and the geophysical information available in the S5P TROPOMI data.



## 2 TROPOMI tropospheric ozone column data

### 2.1 TROPOMI instrument

TROPOspheric Monitoring Instrument (TROPOMI, Veefkind et al. (2012)) is the unique payload on the Copernicus S5P
satellite, the first atmospheric composition mission in the EU Copernicus Earth Observation programme (Ingmann et al., 2012).
TROPOMI was launched into a sun-synchronous low Earth orbit on Friday October 13, 2017. The ascending node of the orbit
crosses the equator at 13:30 solar local time. The four imaging spectrometers of TROPOMI measure the spectral radiance
scattered at nadir from the sunlit part of the atmosphere and the solar spectral irradiance, in the 270–2385 nm wavelength
range at 0.2–0.5 nm resolution. The field of view at nadir produces ground pixels of $5.5 \times 3.5 \, \mathrm{km^2}$ (along$\times$across-track) since
the pixel size switch of August 6, 2019, and of $7 \times 3.5 \, \mathrm{km^2}$ before. The large swath width of 2600 km across-track produces
almost a daily coverage of the global atmosphere. After spectral and radiometric calibration of the Earth radiance and solar
irradiance data (Kleipool et al., 2018; Ludewig et al., 2020), operational data processors retrieve the column abundance of
several atmospheric trace gases related to air quality, climate, stratospheric ozone depletion, UV radiation and environmental
hazards. They retrieve in particular the vertical column amount of ozone and the cloud parameters needed for the computation
of tropospheric ozone by application of the CCD technique described hereafter.

### 2.2 Convective Cloud Differential algorithm

Unlike other TROPOMI atmospheric data products, the tropospheric ozone column data are not retrieved directly from the
radiance data using an inversion scheme. Rather, it is derived from total ozone column and cloud data using the Convective
Cloud Differential approach (CCD), a technique that has been applied successfully to many other sensors such as TOMS,
GOME, SCIAMACHY, OMI and GOME-2 (Ziemke et al., 1998; Valks et al., 2003, 2014; Heue et al., 2016; Leventidou et al.,
2016, 2018).

     The TROPOMI implementation (ESA, 2018) inherits from that used for earlier GOME-type sensors (Valks et al., 2003, 2014;
Heue et al., 2016). The first step consists of selecting ozone columns retrieved by the GODFIT V4 algorithm (Van Roozendael
et al., 2012; Lerot et al., 2014; Garane et al., 2019) over deep convective clouds in the tropical eastern Indian and western
Pacific Oceans (70° E–170° W, 20° S–20° N). Outside this reference region thick, high and highly reflective clouds occur less
frequently. Deep convective clouds are identified by high cloud fraction ($\geq$0.8), high cloud albedo ($\geq$0.8) and low effective
cloud pressure ($\leq$300 hPa), information which is retrieved using the OCRA/ROCINN_CRB algorithm by the UPAS v1 data
processor (Loyola et al., 2018; Compernolle et al., 2020). In a second (standardisation) step, the missing (or redundant) ozone
column between the fixed reference level of 270 hPa and the retrieved effective cloud pressure is added (or subtracted) using
estimates from the McPeters and Labow (2012) ozone profile climatology. The standardised columns are subsequently averaged
over five day windows and over 0.5° latitude bins, then smoothed with a running mean over 2.5° in the latitudinal domain to
reduce the effect of undersampling. These values are reported as the reference stratospheric ozone column (StOC) in the data
products and typically include 1000–2000 TOC pixels. Then (step 3), total ozone columns over clear-sky scenes (CF<0.1)
are averaged over three days in 0.5° latitude by 1° longitude grid cells (on average ∼100 TOC pixels). In the fourth and



final step, a tropospheric ozone column (TrOC) is obtained over each cell by subtracting the corresponding StOC from the cloud-free total column. This explicitly assumes that StOC is uniform over the entire latitude belt and representative of the three day mean state.

## 2.3 Data record and screening

The resulting TROPOMI tropospheric ozone column product is sampled daily and represents clear-sky three day moving
mean ozone column between surface and $270\,\text{hPa}$ around 13:30 over $0.5° \times 1°$ grid cells between $20°$ S–$20°$ N. Hence, the TROPOMI CCD tropospheric column does not cover the uppermost part of the free troposphere nor the tropical transition layer. Data during the commissioning phase (7 November 2017 to 30 April 2018) are not considered here since not publicly released. We used two complete years of TROPOMI tropospheric ozone column data (1 May 2018 to 30 April 2020), corresponding to offline processors V01.01.05–V01.01.08. Differences between versions are negligible and mainly reflect a change in input
TOC or cloud data (ESA, 2020). Figure 1 shows the median and 68% interpercentile of screened TROPOMI TrOC data over the two year period. Features and patterns in these maps are discussed in subsequent sections.

   Only data with a quality assurance value above 0.7 are retained for this analysis, as recommended by the data provider (ESA, 2020). The quality assurance variable combines different quality criteria, primarily the mean quality of the selected cloud-free TOC pixels and several indicators of elevated error in StOC (large zonal variability in reference sector, too few deep convective
clouds, large difference between latitude bands). Additional quality criteria include the sample size and standard deviation of cloud-free TOC pixels and the fraction of negative TrOC in the spatio-temporal window. If the final TrOC value is negative it is overwritten by a fill value. Applying the recommended screening removes about 5–10% of the data in the inner tropics and up to 40–50% close to $20°$ latitude. In the outer tropics, mostly wintertime data are rejected (Fig. S1 in Supplement) because there is a lack of deep convective clouds while the inter-tropical convergence zone (ITCZ) is located in the opposite hemisphere.
The seasonal migration of the ITCZ (Holton and Hakim, 2013) therefore leads to a temporal sampling bias in yearly averaged TrOC data. Only data in the inner tropics ($12°$ S–$10°$ N) exhibit a uniform temporal sampling distribution over a twelve month period (Fig. S1, bottom, white shade). In the outer tropics, the sampling barycentre of annual mean TrOC is located either in February (SH) or in August (NH).

## 2.4 Sources of uncertainty

Errors in the derivation of tropospheric ozone column data originate in the retrieval of total ozone and cloud information, in the validity of underlying assumptions and in the representativeness of the data sample. Potentially the most important source of systematic error in TrOC lies in a difference between systematic error in clear-sky and fully cloudy TOC. While challenging to unveil a cloud dependence of TOC bias in ground-based comparisons, we found evidence suggesting that cloudy TOC data are biased low by (at least) 1% with respect to cloud-free scenes (Sect. 4.4, Fig. 6). Such an error would lead to positive bias
of (at least) $2\,\text{DU}$ assuming that 10% of the total column resides in the troposphere. It should therefore be subject of a further investigation which takes into account errors in the cloud retrieval as well (Compernolle et al., 2020) and potential biases in the ground-based data at high CF.



An important source of random uncertainty in TrOC comes from the assumption that the ozone field in the stratosphere and tropical transition layer is uniform along longitude. Various studies, each using complementary techniques, concluded that this is a reasonable approximation at low latitudes ($<15°$) where observed RMS of stratospheric and TTL ozone within a latitude belt ranges between 2–5 DU (Ziemke et al., 2009b, 2010; Valks et al., 2014; Thompson et al., 2017). In addition to geophysical variability, sampling errors related to the presence/absence of deep convective clouds over the reference region will affect the StOC estimate. Any error in StOC propagates directly to the errors of all TrOC values over the corresponding latitude belt. Sampling effects may play a role too in the set of cloud-free TOC pixels which are used to derive the final TrOC value.

Biases in the retrieval of effective cloud pressure for convective clouds will bias the 270 hPa standardisation step and therefore cause systematic error in StOC. Ozone mixing ratios are generally small over the Indian and Pacific oceans, so it is generally assumed the final impact on TrOC is fairly small. The standardisation step also introduces some random uncertainty since a climatology is used which is representative of the mean state over decadal time scales.

The uncertainty reported in the TROPOMI data files reflects a purely random component and is computed as the standard deviation of the set of estimated TrOC within each spatio-temporal bin. It is therefore the combined result of random measurement uncertainty and random geophysical variability within the bin. However, not included are the random representativeness uncertainty resulting from inhomogeneous spatio-temporal sampling over the bin nor the random uncertainty in the estimate of the StOC reference. In the following, we refer to the reported uncertainty as ex ante uncertainty (von Clarmann, 2006; von Clarmann et al., 2019), they typically range between 1.7–2.1 DU ($\sim$7–13%) and increase somewhat (by at most 0.5 DU) over the South Atlantic Anomaly (Fig. 1 bottom).

## 3 Reference TrOC data records and harmonisation

### 3.1 Ozonesonde

Attached to a small meteorological balloon, ozonesondes measure in situ the abundance of tropospheric and lower stratospheric ozone at the effective vertical resolution of $100 - 200$ m. The instrument consists of an electrochemical concentration cell (ECC) that produces an electrical current proportional to the partial pressure of ozone (Tarasick et al., 2020, and references therein). The ozonesonde is mated to a meteorological radiosonde which provides simultaneous readings of timestamp, GPS geolocation, ambient pressure and temperature. Soundings in the 20° S–20° N belt are generally made weekly or every other week, at nine stations affiliated to NASA's Southern Hemisphere ADditional OZonesonde programme (Thompson et al., 2017, 2019) and the Network for the Detection of Atmospheric Composition Change (NDACC, De Mazière et al. (2018)). Most sites launch their balloons at a fairly fixed local solar time, but the median launch time does vary considerably across the network, between 4:45 and 12:45 (Fig. S5 in Supplement). We considered rapid delivery data from the NOAA and KNMI archives and homogenised version 6 data from the SHADOZ data archive (https://tropo.gsfc.nasa.gov/shadoz).

Ozonesonde profile data are screened as described in Hubert et al. (2016). Additionally, Costa Rica soundings flying through a plume of the nearby Turrialba volcano were discarded, as $SO_2$ interferes strongly with the ECC cell and this causes too low ozone readings over a considerable portion of the troposphere (Morris et al., 2010). After screening, the sonde ozone volume



mixing ratio profiles are integrated using Eq. A1 (see Appendix A) from the first measurement level up to 270 hPa to obtain a tropospheric column over the same vertical range as TROPOMI. The integration method implicitly assumes that TROPOMI TrOC data have full and uniform sensitivity over the entire partial column and no sensitivity at higher levels. In rare cases, the first reported sonde reading occurs more than 100 m above the surface. If the log(pressure) range sensed by the sonde misses 3% or more of the surface–270 hPa range, the sonde flight is discarded.

Over the past few years the ozonesonde community made significant progress in refining the characterisation of errors and in correcting for inhomogeneities in the data records across the network. The latter process references the sonde records to a UV-absorption photometer (Thompson et al., 2019; Tarasick et al., 2020), which, in principle, eliminates biases between sites as well as biases between different periods at a single site. In reality, however, the homogenisation process leaves residual systematic differences of about 5% (or ~1 DU) in the troposphere (Thompson et al., 2017; Tarasick et al., 2019).

In the tropical troposphere, total uncertainty estimates typically lie in the 5–15% range for pressures larger than 270 hPa (Witte et al., 2018; Sterling et al., 2018). However, so far, the error components are rarely separated in the data files which makes it impossible to propagate errors for post-processed sonde data in a correct fashion. The nature of the error (random or systematic) for each domain (vertical, horizontal, time) determines how it propagates through the post-processing routine of the data user. This will be different for the integration into a tropospheric column, for the computation of monthly averages or regional means.

The SHADOZ network pioneers the provision of a more complete error decomposition. Unfortunately, the error due to the background current in the sonde, which dominates the uncertainty budget in the tropical troposphere, is not differentiated from the sensor current error in the V1 uncertainty data files. Both error sources exhibit random behaviour over all domains (vertical, between flights at a site, between sites) except for the background current error which is systematic in the vertical domain. Since other large sources of error also display systematic behaviour in the vertical, we assume that the reported total error is fully correlated between pressure levels. Hence, a (slightly conservative) error on the tropospheric column is obtained by applying Eq. A1 (in Appendix) to the ozone volume mixing ratio error profile. This procedure was also followed by Witte et al. (2018) in their comparison of total column data. We find tropospheric column errors of 0.9–1.5 DU or 5–8%.

Stauffer et al. (2020) reported an unexplained 5–10% drop in stratospheric ozone levels around 2014–2016 at several sites that launch ECC instruments manufactured by the ENSCI corporation. All but two SHADOZ sites (Paramaribo and Kuala Lumpur) used in this work are affected. The cause of the drop is not understood and not included in the published uncertainty budgets (e.g., Witte et al., 2018; Tarasick et al., 2020). It is possibly related to a change in ENSCI instrumentation, although other causal factors are explored as well. But the artificial decline was not noticed in the tropospheric part of the profiles at the impacted sonde sites with the exception of Costa Rica. There, a drop-off occurred early 2016 in the troposphere (Stauffer et al., 2020). Efforts to correct for the recent low bias in Costa Rica sonde data are promising and a revised record is being prepared (Vömel et al., 2020). The comparisons over Costa Rica are therefore not considered for the estimation of TROPOMI bias. Since the breakpoint occurred before the start of the TROPOMI data record, it does not alter estimates of correlation, dispersion or temporal stability of the comparisons. We therefore assume that the drop-off issue can be ignored in our analysis of tropospheric ozone data.



## 3.2 OMI and GOME-2B

OMI and GOME-2B are part of the European family of UV-Visible nadir-viewing spectrometers pioneered in the mid-1990s with ERS-2 GOME, a family of which TROPOMI is the youngest member. OMI resides on the Aura satellite launched in 2004, GOME-2 is part of the MetOp-B platform deployed in 2012. These sensors are in a sun-synchronous low Earth orbit
with an equator crossing local time of 9:30 (GOME-2B, descending node) and 13:45 (OMI, ascending node). As part of the European Space Agency's Climate Change Initiative (CCI), total ozone columns have been retrieved for all European sensors using the GODFIT V4 algorithm (Lerot et al., 2014; Garane et al., 2018) and these were subsequently passed through the CCD algorithm to infer gridded tropospheric ozone columns in the tropical belt (Heue et al., 2016).

Here, we use a slightly modified version of the OMI and GOME-2B CCI data records spanning November 2017 to, re-
spectively, February 2020 and January 2020. The spatio-temporal sampling resolution was aligned to that of TROPOMI to reduce sampling and smoothing mismatch uncertainties in our comparison (see Sect. 4.1). The original OMI and GOME-2B CCI data represent monthly-sampled monthly mean ozone columns between surface and $10\,\mathrm{km}$ for $1.25° \times 2.5°$ cells (latitude $\times$ longitude) between $20°$ S–$20°$ N. The data used here are daily-sampled five-day mean ozone columns between surface and $270\,\mathrm{hPa}$ in $1° \times 2°$ cells. Grid cells and GOME-2B total ozone pixels footprints are of similar magnitude (Table 1), therefore
the East-West overlap is used as a weight in the gridding process for this sensor.

Uncertainties reported in the GOME-2B and OMI data files are computed in the same way as for TROPOMI. Ex ante random uncertainty for OMI TrOC data lies in the $2.5$–$3.1\,\mathrm{DU}$ range, for GOME-2B these are slightly smaller $1.8$–$2.6\,\mathrm{DU}$ but the latter exhibit a much more pronounced peak around the South Atlantic Anomaly ($4.5$–$6.5\,\mathrm{DU}$ compared to $3.5$–$4.5\,\mathrm{DU}$ for OMI). Systematic errors were not propagated through the measurement process. Instead, systematic measurement uncertainty
is indirectly probed by comparison to ground-based and satellite data records, an approach that is followed here as well. Heue et al. (2016) report a $\sim$3 DU low bias of GOME-2B with respect to OMI, the authors suggest this may be caused by the different local measurement time in the presence of a diurnal cycle in tropospheric ozone. We discuss this further in Sect. 4.4.

## 3.3 Co-location with TROPOMI

Table 1 summarises the sampling and smoothing properties of the considered TrOC data records. Thanks to the excellent
spatio-temporal coverage of TROPOMI TrOC data, tight co-location windows can be used without loss of comparison pairs. Ozonesonde measurements at a given time and from a given geolocation are compared to the corresponding satellite space-time cells, which limits the mismatch between the TROPOMI grid cell centre and the ozonesonde data to a maximum of $\pm1.5$ days, $\pm0.25°$ in latitude and $\pm0.5°$ in longitude. Sonde TrOC data are averaged whenever multiple launches occur at a single station in TROPOMI's three-day window, although this is very rare.
Satellite intercomparisons are carried out on the coarser horizontal grid of OMI and GOME-2B. For each OMI or GOME-2B spatial cell, four TROPOMI cells are averaged with uniform weights. The reported TROPOMI uncertainties are assumed uncorrelated and propagated accordingly. TROPOMI data were not averaged over the time domain which leaves a difference in temporal smoothing (three versus five days).





**Table 1.** Sampling and smoothing properties of tropical tropospheric ozone column data records (surface–270 hPa, 20° S–20° N). The horizontal column displays latitude by longitude for the tropospheric column, and along- by across-track pixel footprint at nadir of the total ozone columns used by the CCD algorithm. Data records are ordered by smoothing area.

| Sensor | Horizontal | Time (local solar) | Other |
|---|---|---|---|
| ozonesonde | flight path, drift $< 5$–15 km, nine sites | 30 min ascent, sampled 1–2x/month at 2:20–15:20 | all weather |
| TROPOMI | mean over $0.5° \times 1°$ using $5.5 \times 3.5\,\mathrm{km}^2$ pixels[†] | 3-day mean, sampled daily at 13:30 | cloud-free |
| OMI | mean over $1° \times 2°$ using $13 \times 24\,\mathrm{km}^2$ pixels | 5-day mean, sampled daily at 13:45 | cloud-free |
| GOME-2B | mean over $1° \times 2°$ using $80 \times 40\,\mathrm{km}^2$ pixels | 5-day mean, sampled daily at 9:30 | cloud-free |

[†] TROPOMI pixel footprints were slightly larger before 6 August 2019, 7 km along-track.

## 4  Comparison to ozonesondes and satellites

The first part of our analysis consists of the comparison of TROPOMI TrOC data to co-located measurements by ozonesonde and satellite instruments. Its purpose is to derive statistical indicators of TROPOMI data quality, such as systematic error or random uncertainty, and to study their spatio-temporal structure. Robust estimators are used to limit the impact of outliers in the (relatively) short and sparsely sampled co-location data set. In subsequent sections, we take a closer look at patterns in the TrOC maps in order to infer additional uncertainties caused by sampling (Sect. 5) and to verify the ability of TROPOMI to

record known geophysical patterns (Sect. 6). Before going into the presentation of TROPOMI TrOC comparison results we highlight the importance of confounding factors in the interpretation of these comparisons.

### 4.1  Comparison error budget

#### 4.1.1  General

The geophysical interpretation of atmospheric measurements requires proper consideration of the random and systematic mea-

surement uncertainty, but also understanding of the location and extent of the probed air mass. Indeed, a measurement of the geophysical state at a point in 4D space-time by a hypothetical, perfectly accurate instrument (i.e., zero measurement error) will deviate from the true state value since the information actually probed by the measurement process will be smeared out and/or displaced from the targeted location. Such a deviation is often referred to as representativeness (or mismatch) error, and it exhibits random and/or systematic behaviour depending on the spatio-temporal structure in the geophysical field and the

sampling and smoothing properties of the measurement system.

A primary objective in the validation of data record $X$ is to quantify its systematic ($\beta_X$) and random ($\sigma_{\epsilon_X}$) measurement uncertainty. The usual approach is to compare the data record to measurements by another instrument $Y$. In order, then, to infer the measurement uncertainty ($\beta_X, \sigma_{\epsilon_X}$) from a set of differences $\Delta = X - Y$ we must know several nuisance parameters : systematic ($\beta_Y$) and random ($\sigma_{\epsilon_Y}$) measurement uncertainty of $Y$, and the systematic ($\beta_{repr}$) and random ($\sigma_{\epsilon_{repr}}$) uncertainty

due to the different representativeness of the two measurements. These nuisance parameters, furthermore, often depend on time





and location. Their magnitude can be similar to the measurement uncertainty of interest and they are often difficult to quantify. In some cases, however, representative observations operators applied to modelled fields allow to simulate - in an Observing System Simulation Experiment (OSSE) - the measurement process sufficiently well to close the comparison error budget (e.g., Verhoelst et al., 2015).

A common approach is to reduce the uncertainty from nuisance parameters as much as possible, e.g., by using a well-calibrated data record as a reference and/or by harmonising the representation of the data records (Keppens et al., 2019). Systematic and random measurement uncertainties for sonde, OMI and GOME-2B are discussed in Sect. 3. In general, the estimation of measurement uncertainty is more intricate than the measurement itself. As a result, the reported uncertainties are often first order approximations although, at times, they fail to include important, poorly understood sources of error. It is

therefore good practice not to use these uncertainty estimates blindly.

Section 3 also describes how vertically resolved ozonesonde data are integrated to a partial column, how the data records are co-located in space and time to reduce errors from differences in sampling, and, how the spatio-temporal resolution of the best resolved data is downgraded to that of the coarser resolved record to reduce errors from differences in horizontal smoothing. State-of-the-art models resolve global tropospheric ozone at best at the scale of $100 \times 100 \, \mathrm{km}^2$ (Young et al., 2018), which

is too coarse to simulate the representativeness errors due to geophysical variability within the co-location window. Hence, a detailed closure of the comparison error budget for tropospheric ozone records is currently out of reach (see also Tarasick et al., 2019). In the following, we resort to a qualitative discussion of representativeness uncertainty and its decomposition in the temporal, horizontal and vertical domain.

### 4.1.2  Systematic representativeness uncertainty

A classical source of systematic component in the representativeness error is the difference in local measurement time in the presence of a diurnal cycle. Not much is known about the strength of the diurnal cycle of ozone in the tropical troposphere. Measurements around the globe show clear evidence of changes in surface ozone with local time, with amplitude and timing of minimum/maximum ozone depending on parameters like the strength of solar radiation driving photochemical reactions and the presence of precursor emissions in the vicinity of the site (e.g., Tarasova et al., 2007; Schultz et al., 2017). Much

fewer observations are available in the boundary layer and, especially, the free troposphere. Thompson et al. (2014, Fig. 4) describe a correlation between local time and ozone mixing ratios in the boundary layer from soundings at the subtropical Irene station (25.9° S, 28.2° E, South Africa) between 1999–2007. Petetin et al. (2016) characterised the diurnal variation of ozone close to the surface over Frankfurt (50.0° N, 8.6° E, Germany) in MOZAIC-IAGOS aircraft data. Both studies conclude there is no evidence for a cycle in the free troposphere. Annually averaged ozone columns between surface and 300 hPa

over Frankfurt are at most ∼1.1 DU larger at noon (12–18 h) than during nighttime (21–09 h). The mismatch in measurement time with respect to TROPOMI is negligible for OMI, but not for sonde or GOME-2B (Table 1, Fig. S5). Assuming that the diurnal cycle in the tropics resembles the one over Frankfurt, we expect a systematic error of about +1 DU in comparisons of TROPOMI minus GOME-2B or ozonesonde at most sites. Mismatch error would be smaller (∼0.5 DU, or less) in comparisons to Natal, Paramaribo (from January 2019 onward) and Ascension Island sondes since these sites launch around noon, closer





to TROPOMI overpass time (Fig. S5). Paramaribo flights prior to 2019 were launched around 9:45 local solar time, leading to larger mismatch with TROPOMI.

Another source of systematic representativeness error can be the difference in vertical smoothing of the tropospheric profile by the ozonesonde and by TROPOMI. Sonde data are integrated assuming that TROPOMI TrOC data have uniform, full sensitivity between surface and 270 hPa and no sensitivity elsewhere. In reality, TROPOMI total ozone column retrievals have
reduced sensitivity close to the surface and increased sensitivity above clouds, as shown by the associated averaging kernels (ESA, 2018). It is outside the scope of this work to study how vertical sensitivity for single TOC pixels in cloud-free and cloudy scenes propagates into a regional multi-day mean tropospheric column. However, it is clear that the magnitude of this effect will depend on the actual ozone profile and on the cloud coverage and properties and, therefore, that this systematic error will have spatio-temporal structure.

### 4.1.3   Random representativeness uncertainty

Spatial correlation length in the troposphere is about 500 km, larger than the grid cells in the satellite products and much larger than the horizontal distance travelled by ozonesonde. Temporal correlation length is about 1.5–3.5 days, which is close to or smaller than the averaging window used by the CCD algorithm (Table 1). These correlation lengths were estimated by Liu et al. (2009) from an analysis of ozonesonde data in Europe and the USA. Assuming these hold for tropical conditions as well,
we expect that temporal smoothing difference is the main contributor to random representativeness error.

Satellite TrOC data are smoothed over a much larger time window and region than ozonesonde data (Table 1). Hence, random representativeness uncertainty will be significantly larger in satellite to sonde comparisons. Again, state-of-the-art models are too coarse to simulate their magnitude. Differences in smoothing between satellite sensors are much smaller and therefore the random representativeness errors in the satellite intercomparisons will be smaller. Smallest random uncertainty is
expected in the comparison of TROPOMI and OMI. For these two sounders ground pixel footprints of total ozone retrievals are much smaller than the cell size of the TrOC data record, therefore the error due to horizontal smoothing differences should be negligible. GOME-2B pixels, on the other hand, are resolved fairly close ($80 \times 40 \, \text{km}^2$) to the TrOC cell size (1° by 2°). This effectively leads to horizontal smearing in GOME-2B TrOC and a larger random uncertainty in TROPOMI-GOME-2B comparisons.

## 4.2   Temporal correlation

Figure 2 shows time series of ozonesonde (red) and TROPOMI (black) TrOC data over nine SHADOZ stations. Temporal co-location is neglected in this figure to highlight the variability seen by TROPOMI over the full range of time scales. Both records show a coherent picture of the known large-scale spatio-temporal patterns in tropical tropospheric ozone: zonal wave-one, annual cycle, biomass burning and convective activity. We return to the verification of these geophysical signals in Sect. 6.
In the rest of Sect. 4 we consider TrOC data that are spatially and temporally co-located.

Pearson correlation coefficients are estimated using the skipped correlation measure $r$ which is robust against outliers (Wilcox, 2004). TROPOMI correlates reasonably well with spatio-temporally co-located ozonesonde data, $r$ ranges between





45–75% over the network and averages at 61% (Fig. 4). Correlation with satellite data records is (much) stronger, between 60% and 95%, and it traces the contours of variability in the TrOC field (Fig. S2 in Supplement). In regions with $\sigma_{\mathrm{TrOC}} > 6\,\mathrm{DU}$ over

the course of a year (e.g., South America), the correlation is more than 80%. Correlation is weaker, less than 70%, in regions of low atmospheric variability ($\sigma_{\mathrm{TrOC}} < 4\,\mathrm{DU}$, e.g., equatorial western Pacific), also because TrOC values are generally lower at these locations and hence the relative importance of uncorrelated random measurement error is larger.

The different correlation strength of TROPOMI with the three reference data records reflect, in part, the difference in representativeness of the data records. TROPOMI-OMI correlations are 2-4% higher than for GOME-2B, since the hori-

zontal smoothing of the latter sensor differs more from TROPOMI than OMI's does. The weaker correlation TROPOMI-ozonesonde coincides with a large difference in spatio-temporal smoothing. Overall, the reasonably high correlations indicate that TROPOMI TrOC data capture the general temporal variability observed by the other data records.

## 4.3 Temporal stability

Comparisons of TROPOMI to sonde and satellite display stable statistical dispersion over the two year record. However, the

mean level in the satellite intercomparisons exhibits a clear modulation with 0.7–1.2 DU amplitude across the entire tropical belt (Fig. 3). Such a small effect can currently not be seen in the sonde analysis but may be with longer time series (Fig. 4). Figure 3 suggests a seasonal pattern in the anomaly of TROPOMI minus OMI (top) or GOME-2B (bottom) with respect to the mean value over the entire period. Details of the bias anomaly pattern in both satellite intercomparisons are reasonably consistent. A minimum occurred in September–January 2018, a maximum during March–July 2019. The phase of the modulation

is uniform across latitude in the GOME-2B results, but varies by ∼3 months in the OMI comparisons. Also, the modulation amplitude in the latter analysis is 0.3–0.5 DU larger. Longer time series are needed to confirm whether these changes in TROPOMI bias persist as a cyclic pattern, whether it was merely a single episode, but also, whether a coherent seasonal bias in GOME-2B and OMI data can be ruled out.

Two additional temporal features in the ground-based comparisons are noteworthy. The most striking is that TROPOMI

observes 5–10 DU higher TrOC values than the Paramaribo ozonesonde during the 2018 and 2019 biomass burning seasons (Fig. 4). Over this site, the rapid increase (July) and decline (November) in TROPOMI data is not, or only weakly, present in the ozonesonde record. This discrepancy leads to the low correlation of 45% at this site. At other sites around the Atlantic basin (Ascension Island, Nairobi and Costa Rica) TROPOMI bias increases over the July-September 2018 period, like at Paramaribo, but the pattern does not persist as clearly in 2019. A short high-bias episode at Natal and Ascension Island occurred in October–

November 2019, coinciding with lower than usual (5–10 DU) sonde readings. Figures S3 and S4 in Supplement show the same temporal pattern in the comparisons of both OMI and GOME-2B to ozonesonde, implying that it is not related solely to TROPOMI. Further study is needed to understand whether this simultaneous temporal change in bias around the Atlantic has a common source for the three satellite sensors (e.g., instrument design, calibration, total ozone retrieval, tropospheric ozone derivation through CCD), whether the ozonesonde records were simultaneously affected by an offset in the measurements

during 2018, or, whether the difference in vertical smoothing of the tropospheric column by sonde and satellite has a larger impact during high ozone conditions.





A second anomalous event occurred during the last months of 2019 in the tropical South Pacific, coinciding with a period of widespread bush fires across Australia. Three ozonesonde flights in October-December 2019 at Suva recorded higher TrOC values than usual, while co-located TROPOMI data did not. This leads to a temporary low bias of TROPOMI with respect to

sonde that deviates by 5–15 DU from the rest of the time series (Fig. 4, bottom centre). Comparisons at Samoa show a similar but less pronounced dip in December 2019 (Fig. 4, bottom right). Inspection of maps of TROPOMI CO, a tracer for smoke plumes, around the ozone soundings did not reveal clear evidence of a link with the bush fires ∼3000–4000 km away, in South East Australia.

## 4.4  Bias

The 50% quantile of the ΔTrOC distribution is used here as robust estimator of TROPOMI bias with respect to the other data records. This bias estimate is an indirect probe of TROPOMI's systematic measurement error since other systematic error terms are at play (Sect. 4.1.2). The Costa Rica comparisons are not included in this part of the analysis due to a known low bias (Sect. 3.1). Figures 4 and 5 (bottom) show the bias of TROPOMI with respect to ozonesonde data at each SHADOZ site. TROPOMI generally reports higher values than the ozonesondes, by 2.3±1.9 DU (or 11.2±9.0%) when averaged over

the network. The error bar represents the statistical dispersion (1$\sigma$) of the bias estimates over the ground-based network. A closer look suggests three clusters in the bias estimates: three sites (Suva, Kuala Lumpur and Samoa) where TROPOMI bias is less than ∼1 DU (4–6%), Natal with a bias of +2 DU (9%) and a group of four (Hilo, Paramaribo, Ascension Island and Nairobi) with a bias around +4 DU (14–22%). A similar clustering is seen in the ground-based TrOC comparisons to OMI and GOME-2B (Figs. S3 and S4). Differences in sampling of the diurnal cycle can not explain this clustering. Instead, it is more

likely that residual instrument-related biases of ∼15% exist between the ozonesonde stations. This hypothesis is supported by independent total ozone column comparisons between sonde and satellite (OMI and Suomi-NPP OMPS-LP) which exhibit a similar grouping (Stauffer et al., 2020).

TROPOMI also overestimates GOME-2B TrOC data across the entire tropical belt, by 2.3±0.6 DU (+13.2±5.2%) on average. Again, the error bar represents the statistical dispersion (1$\sigma$) of the bias estimates over the tropical belt. The agreement

with OMI is much better, TROPOMI underestimating OMI data by 0.1±0.7 DU (-0.3±3.2%). The bias display spatial structure that differs from one instrument to another. When compared to OMI, TROPOMI biases exhibit a marked meridian and zonal dependence (Fig. 5, top). Biases in the Southern tropics are +(0.2–0.4) DU and -(0.6–1.0) DU in the North. Also, biases are larger around the Greenwich meridian (-0.8 DU) than around the dateline (±0.2 DU). It is unclear whether the latter finding is related to the fact that the reference stratospheric ozone column used in the CCD retrieval is derived in the Pacific sector.

In any case, such a zonal structure is not seen in the TROPOMI comparisons with GOME-2B. The meridian dependence in the latter has the same sign but is much weaker, with only 0.2–0.4 DU difference between the Northern and Southern tropics. The GOME-2B comparisons also show signs of change in bias around coastlines or over high elevation terrain. This is not seen in the OMI results, so it may be a systematic effect in the GOME-2B data or due to the larger difference in horizontal smoothing. Interestingly, the sonde comparisons show a smaller bias in the reference region (Suva, Samoa, Kuala Lumpur)





like OMI. However, the 1.9 DU scatter in the sonde-based bias estimates and the sparsity of the network challenge a third and more independent view on these spatial patterns.

  The TROPOMI bias and its structures could be the result of several systematic terms, such as sampling mismatch of the diurnal cycle, biases in TROPOMI total ozone and cloud retrievals, and vertical smoothing issues. It is not straightforward to substantiate these potential sources of biases. In Sect. 4.1 we estimated that the difference in measurement time by the

ozonesondes and GOME-2B with respect to TROPOMI could contribute 0.5–1 DU to the bias results. Correcting for the diurnal cycle effect would reduce the observed TROPOMI bias relative to sonde from 2.3 DU to about 1.3–1.8 DU and with respect to GOME-2B bias from 2.3 DU to about 1.3 DU.

  Other potential sources of bias are the systematic errors affecting the TROPOMI total ozone column and TROPOMI cloud data used by the CCD algorithm. A global, absolute offset in total ozone column (TOC) would not affect TrOC data since this

offset would be removed by the CCD approach: indeed, TrOC is essentially the difference between TOC over cloud-free scenes and TOC over very cloudy, high cloud scenes. However, a cloud dependence of the TOC bias could produce a systematic bias in TrOC data. Ground-based validation of TROPOMI TOC in Garane et al. (2019) concludes that, on average, there is no clear dependence of TROPOMI TOC bias on cloud parameters, but this study does not specify to what level of confidence this dependence can be excluded. Therefore, specific comparisons have been undertaken here on two years of TROPOMI TOC

(OFFL, V01.01.01-01.01.08) with respect to the Brewer and SAOZ network measurements, which suggest a dependence of TOC bias on the TROPOMI cloud fraction (Fig. 6). TROPOMI TOC over very cloudy scenes (CF>80%) appears to be biased low by about 1% (Brewer comparisons) to 2.5% (SAOZ comparisons) with respect to TOC over cloud-free scenes (CF<10%). On the one hand, several elements suggest that this dependence is a real feature of the satellite TOC: (a) the dependence is present across the entire CF range (i.e., not only at very high CF where uncertainties in the ground-based data are also

significant), (b) the Brewer and SAOZ measurements are largely independent and have different sources of errors, and, (c) the networks have a different geographical sampling. The slightly lower CF contrast in the Brewer results is in line with findings by Zhao et al. (2019). On the other hand, there have been reports on positive biases at high CF in both SAOZ and Brewer (zenith-sky) measurements (Zhao et al., 2019; Fioletov et al., 2008, respectively), but these were based on comparisons to reanalyses and satellite data sets, and they may therefore just represent an inverse interpretation of the same CF dependence in

the comparison results. Assuming a similar (and real) CF dependence in the TOC subsets used by the CCD algorithm, then the 1–2.5% low bias of cloudy TOC with respect to cloud-free TOC should lead to a ∼2–5 DU positive bias of the tropospheric ozone column. Since TrOC are very sensitive to a CF dependence in TOC, the latter should be investigated in further detail and better constrained, also for the OMI and GOME-2B TOC data records since seemingly affected by a similar CF dependence (T. Verhoelst, private communication).

A second cloud-related effect comes from a bias in retrieved effective cloud height. Retrievals of cloud height by the TROPOMI ROCINN CRB algorithm tend to be biased low (Compernolle et al., 2020), which leads to too high TrOC values through the normalisation of the above cloud ozone column to the 270 hPa reference level. It is challenging to validate cloud data quality since ground-based instruments perceive clouds in a different way than TROPOMI. Compernolle et al.





(2020) report a negative bias in cloud height of 0.5–1.5 km with respect to CLOUDNET cloud mean height data which is then
estimated to invoke a positive bias in TrOC data of up to ∼0.5 DU.

Finally, a difference in vertical smoothing will introduce systematic mismatch error in the satellite-sonde comparisons as
well. Our assumption that satellite CCD TrOC data have uniform sensitivity over the tropospheric column is a first-order
approximation. In reality, the retrieval is undersensitive to ozone close to the surface and slightly oversensitive to ozone just
above the cloud top. Removing this bias requires the provision of a reliable vertical averaging kernel for the CCD TrOC data,
which is currently under investigation by the S5P MPC. This effect plays no role in the satellite comparisons since the vertical
smoothing properties of the sensors are similar.

### 4.5 Dispersion in pairwise comparisons

Figures 4 and 7 show the dispersion in comparisons of TROPOMI TrOC to co-located data from the three reference data
records. Dispersion is here estimated as half the interval between the 16% and 84% quantiles of the ΔTrOC distribution,
which corresponds to one standard deviation of a Gaussian distribution. At individual sonde stations the dispersion esti-
mates range between 3.0–7.4 DU (or 12–34%), while mean dispersion and its standard deviation across the network equal
4.6±1.3 DU (23±7%). Dispersion for TROPOMI-satellite intercomparisons is generally considerably smaller than the sonde
results: 2.6±0.5 DU (14±5%) w.r.t. OMI and 2.9±0.6 DU (19±7%) w.r.t. GOME-2B. Furthermore, a clear meridian depen-
dence is found in the satellite intercomparisons, with minimal dispersion at the equator and (more than) 1 DU larger values
in the outer tropics. Such a dependence is less clear, but at least not inconsistent, in the ozonesonde comparisons. Also, satel-
lite comparisons show an interesting contrast between dispersion over land or ocean in some regions, especially over South
America and Central Africa. The effect is strongest for TROPOMI-GOME-2B.

Differences in smoothing and sampling by the sensors explain the observed difference in dispersion values and their spa-
tial structure (Sect. 4.1.3). Dispersion values in the GOME-2B comparisons are larger (by 0.2–0.4 DU) than for the OMI
comparisons, most likely due to the larger footprints of the GOME-2B total ozone retrievals. Dispersion is even larger in
the ozonesonde analysis since virtually point-like sonde TrOC measurements are compared to space-time averaged TROPOMI
data. In locations where the tropospheric ozone field is more homogeneous in time and space, e.g. around the equator, mismatch
between sensors will not contribute as much to their comparison than where variability is larger, e.g. in the outer tropics. This
explains (at least partially) the meridian dependence of dispersion found in all analyses and further highlights the challenge of
assessing TROPOMI random uncertainty from pair-wise comparisons.

### 4.6 Random measurement uncertainty from triple co-locations

To overcome the limitations in pairwise comparisons, such as the one faced in previous section, Stoffelen (1998) proposed
to analyse co-locations of three data records $\{X, Y, Z\}$. Under certain assumptions, the triple co-location technique allows to
estimate random measurement uncertainty for each of these data sets $\{\sigma_{\epsilon_X}, \sigma_{\epsilon_Y}, \sigma_{\epsilon_Z}\}$, and none of these should have superior
calibration with respect to the others. The squared random measurement uncertainty, i.e, the variance of the random measure-
ment errors, of data record $X$ can be derived from a combination of variance and pair-wise covariances of the measurement



triplets

$$\sigma_{\epsilon_X}^2 = \sigma_X^2 - \frac{\sigma_{XY}\,\sigma_{XZ}}{\sigma_{YZ}}, \tag{1}$$

under following assumptions: (1) the measurement is a linear function of the true signal with additive zero-mean random mea-
surement noise, (2) measurement errors and true signal are stationary, (3) measurement errors and true signal are independent,
and, (4) measurement errors $\epsilon_X$, $\epsilon_Y$ and $\epsilon_Z$ are independent (McColl et al., 2014; Gruber et al., 2016). The last assumption is
generally the more important one, but also often difficult to validate. For instance, if one sensor has a coarser spatio-temporal
resolution than the other sensors, then it will miss any small-scale geophysical variability picked up by the higher resolution
pair. This small-scale geophysical signal can be seen as a representation error with respect to the lower resolution perspective
of the same state. Since these errors are correlated for the high resolution pair a representation error variance term should be
added to Eq. 1. Generally, this term is challenging to quantify and therefore often neglected. In contrast, if one sensor has a
higher resolution than the other pair, small-scale variability is picked up by just one instrument and the larger scale variability
by all three. Hence, all errors will remain uncorrelated and the representation term vanishes (Vogelzang and Stoffelen, 2012).

A second useful metric accessible through TC is the variance of the geophysical signal $\theta_X$ as measured by sensor $X$,
$\sigma_{\theta_X}^2 = \sigma_X^2 - \sigma_{\epsilon_X}^2$, which scales quadratically with the multiplicative systematic error in the measurement process. $\sigma_{\theta_X}^2$ can be
further related to the variance of the random measurement errors, to obtain a signal-to-noise ratio (SNR). In regions of low
atmospheric variability (e.g., in the equatorial Indian and western Pacific Oceans, Fig. 1 centre) even a small measurement
uncertainty may be too high to pick up the geophysical variability of interest, while higher natural variability at other locations
eases requirements on measurement uncertainty. Here, we rewrite the signal-to-noise ratio (in decibel units) of data record $X$
from Gruber et al. (2016, Eq. 14) in terms of its variance and its error variance

$$\mathrm{SNR}_X = 10\log_{10}\left[\frac{\sigma_X^2 - \sigma_{\epsilon_X}^2}{\sigma_{\epsilon_X}^2}\right]. \tag{2}$$

The three metrics derived through triple co-location ($\sigma_{\epsilon_X}$, $\sigma_{\theta_X}$ and $\mathrm{SNR}_X$) provide complementary information about the
quality of data record $X$ (Gruber et al., 2016). Corresponding estimates for data records $Y$ and $Z$ are obtained by appropriate
permutation of $X$, $Y$ and $Z$.

We perform an analysis of co-located TrOC triplets TROPOMI, OMI and GOME-2B . Prior to the calculation of the covari-
ance matrices non-geophysical outliers in the data records were removed with a Hampel identifier. We argue that representation
errors due to differences in spatial and temporal smoothing of the true TrOC field are negligible. Error cross-correlations in
the spatial domain may occur for the TROPOMI-OMI pair since the resolution of GOME-2B total ozone column retrievals is
markedly lower. However, these errors will be diluted by the temporal smoothing applied by the CCD algorithm. In addition,
representation errors in the temporal domain are negligible since TROPOMI resolution (3 days) is finer than for its two prede-
cessors (both 5 days). We therefore neglect error correlations due to differences in spatio-temporal resolution and apply Eq. 1
to estimate random measurement uncertainty.

Figure 8 shows estimates of random measurement uncertainty (ex post) and these are compared to the mean value of the
measurement uncertainty reported in the data files (ex ante). Ex post uncertainty estimates for TROPOMI (top right) increase





from equator (1.5 DU) to the outer tropics (2.2–2.7 DU). Spatial structure and magnitude of the ex ante uncertainty is fairly
similar. Reported random uncertainties are slightly conservative in the Southern tropics, but not more than 0.4 DU. Overall,
the reasonable agreement between ex ante and ex post uncertainties lends confidence that the uncertainty values reported in the
TROPOMI data files are indeed realistic.

TROPOMI ex post uncertainty is notably smaller than the estimates for OMI (1.9–2.6 DU) and GOME-2B (1.9–3.0 DU),
even though TROPOMI grid cells are four times smaller. The difference in ex post estimates is clearly larger over the South
Atlantic Anomaly, especially for GOME-2B. The better performance of TROPOMI compared to its predecessors is also clear
from the signal-to-noise ratio estimates (Fig. 9). TROPOMI SNR values are generally 1–1.5 dB and 2–3 dB larger than, respec-
tively, OMI and GOME-2B. The spatial structure of SNR is similar for all three instruments: maximal over the southeastern
Pacific, South America and the Atlantic basin, and minimal over the Indian and western Pacific Oceans in the innermost tropics
(coinciding with a minimum in natural variability of TrOC). As a result of the larger measurement uncertainties, signal-to-noise
ratios for GOME-2B and OMI drop over the South Atlantic Anomaly region with respect to TROPOMI.

Figure 8 (centre right) also shows that OMI ex ante uncertainties are overestimated by 0.3–1.0 DU over the entire tropical
belt. Furthermore, sharp peaks were noticed in both OMI ex ante and ex post uncertainty aligned with the orbital track (not
shown here). These are likely due to the row anomaly in OMI Level-1 radiance data, efforts are ongoing to remove these from
the TrOC data. Reported GOME-2B uncertainties, on the other hand, are generally fairly realistic (within 0.5 DU from ex post)
except over the South Atlantic Anomaly where ex ante values are clearly too high, by more than 2 DU (this causes the larger
spread in the Southern Hemisphere in Fig. 8 (bottom right)). The GOME-2B results also hint at a maximal contrast in ex ante
versus ex post uncertainty (i.e., optimistic versus conservative) aligned with coastlines (not shown here).

## 5 Sampling errors at small scales

Some applications may require TrOC data at the finest possible spatio-temporal resolution, in which case random errors can
not be simply averaged out. However, two sources of non-geophysical random uncertainty in TROPOMI data can be (partially)
dealt with, or, at the least, should be considered in the interpretation of analysis results. Both types of error originate in the
spatio-temporal sampling pattern resulting from the interplay between TROPOMI's orbit and cloud coverage. These sampling
errors are correlated at small scales and can dominate the total error. The first type of sampling error leads to a banded structure
in TrOC (0.5–1 DU) along latitude which gradually moves North or South over several days. The second type of error can be
larger (up to 5 DU and more) leading to very localised artificial gradients in time and space.

### 5.1 Sampling of deep convective cloud StOC scenes

Banded structure in the latitude domain can be noted in quite a few TrOC maps, especially in the outer tropics. Such bands
appear where the reference stratospheric ozone column (StOC) has increased error, since this reference is used to infer TrOC
for the entire band (Sect 2.2). Increased error in StOC will, e.g., result from a limited sample of deep convective clouds over the





Pacific reference sector in the five day averaging period. The location of these high, opaque clouds changes over time thereby leading to spatio-temporal correlations in the errors at small scales.

One way to estimate StOC sampling errors is to consider the difference in StOC between consecutive 0.5° latitude bands (referred to as ΔStOC), since this should be a fairly smooth function at small scales. Figure 10 illustrates ΔStOC values of
about 2.5 DU close to 10° S (top right) which leave a visible imprint on the TrOC map of 21 June 2019 (top left). Oscillations in ΔStOC are clear for this particular case but these are, in fact, seen in most maps. The bottom left panel proves this point, as it shows the difference between ΔStOC and the latitude- and time-smoothed ΔStOC field (±2.5° by ±1 day). The smooth field acts as an approximation of the unknown true state and hence the difference can be interpreted as the error in ΔStOC. We find that the StOC-sampling error is strongly correlated in latitude and time across the entire tropical belt, oscillations in latitude
exhibit a period of 2–3° and these structures often persist over 1–2 weeks. These scales are larger than the averaging window (0.5° and five days) used to derive StOC. In Sect. 6.3 we describe patterns in TrOC with a 1–2 week period as well, which are reminiscent of Kelvin wave activity. It is therefore plausible that the described StOC error pattern is in fact geophysical variability. Larger errors are found in the outer tropics, especially during wintertime, which is when the ITCZ is located in the opposite hemisphere. About 10% of the TROPOMI data at latitudes larger than 10° have a StOC sampling error larger than
0.6–0.8 DU (Fig. 10, bottom right, thick red line). In contrast, in the inner tropics, such large errors are found in just 1–2% of the data (thin green and red lines). The mean StOC sampling error is less than 0.05 DU so this effect does not contribute to TROPOMI systematic error.

TROPOMI users interested in fine scale TrOC patterns are strongly advised to consider fine scale structure of StOC as well. This may help them reduce the impact of the StOC sampling error on TrOC data, or, provide the contextual information to better
interpret TrOC patterns. Also, if users choose to relax the QA screening threshold then doing so will introduce considerably more wintertime data in the outer tropics. But this will come at the cost of additional banded structures of higher amplitudes than reported here, due to the much larger corresponding StOC sampling errors.

## 5.2 Sampling of cloud free TOC scenes

A second type of sampling error can have much larger magnitude, but is also more localised in time and space. The CCD
algorithm considers total ozone column (TOC) between surface and top of atmosphere over cloud free scenes, which are then averaged over three days in 0.5° latitude by 1° longitude cells. Sampling in time depends on the location of the clouds, which introduces inhomogeneity. In some cases, cloud free TOC time stamps in neighbouring cells will have a very different barycentre. In conjunction with variability in the true TrOC field over the scale of the three day time window the obtained TrOC may differ by several DU. This TOC sampling effect is often noticed in sequences of TrOC maps, as a localised front
of unnatural changes in TrOC oriented along and propagating with the set of TROPOMI orbits during the three day window.

We infer estimates of this TOC sampling error by collecting the sampling time of all quality-controlled, cloud free TOC values used by the CCD algorithm for a given TrOC map. For each spatial cell these time stamps are averaged and referenced to the central time of the TrOC map. Figure 11 illustrates the temporal inhomogeneity in the sampled cloud free scenes over the Central Pacific used for the map of 20 January 2020. Mean sampling time for nearby cells differs by the maximum possible



amount, three days (top left). This difference leaves an imprint on the TrOC map (bottom left) which becomes especially clear when the anomaly of TrOC is considered with respect to the 1-week smoothed TrOC field (top right). The colour scale shows that TrOC anomalies of neighbouring cells differs by ∼5 DU. The spatial structure of the TrOC anomaly field follows that of the mean sampling time (contours), which corroborates the causal relationship. It is difficult to characterise TOC sampling error, since these need identification on a case by case basis. Inspection of a number of TrOC maps show that the effect is

visible in many maps, at random locations and of varying magnitude. Errors are often 5 DU, or more, so TOC sampling error dominates the total error budget at these locations.

TROPOMI users should understand this type of error, to improve their interpretation of TROPOMI TrOC patterns at the finest scales. A reasonable indicator of increased TOC sampling error is a difference in TrOC anomaly for neighbouring cells in conjunction with a difference in mean sampling time. The latter information is scheduled to be part a future update of the

TROPOMI processor.

## 6 Verification of geophysical information

In the last Section of this paper we explore how TROPOMI captures known signals and patterns in the tropospheric ozone field. This analysis acts as a verification of the ability of the instrument to detect geophysical signals of interest and as a demonstration that it outperforms its predecessors.

### 6.1 Spatial structures

Figure 1 (top) shows median TROPOMI TrOC over two years (1 May 2018 – 30 April 2020). The well known zonal wave-one structure appears clearly, with elevated columns over the Atlantic basin (due to lightning and biomass burning) and depleted levels over the Pacific (due to strong convection in combination with the Walker Circulation) (Thompson et al., 2003; Martin et al., 2007; Sauvage et al., 2007). TROPOMI observes a maximum median TrOC of 33.6 DU (12.25° S, 3.5° E) and a minimum

of 10.0 DU (2.25° N, 155.5° W) resulting in a 23.6 DU peak-to-trough difference (often referred to as the wave-one amplitude). Location and amplitude of the wave-one in two year averaged OMI (24.0 DU) and GOME-2B (22.4 DU) data are similar. Thompson et al. (2017) inferred a smaller wave-one amplitude of ∼14 DU from two decades of SHADOZ ozonesonde data integrated between surface and tropopause. When TROPOMI data are subsampled to the locations of the SHADOZ sites, a more comparable amplitude of 15.2 DU is obtained, giving evidence that the lack of stations in the deep TrOC trough

in the western Pacific is responsible for this smaller wave-one amplitude estimate. Zonal wave-one amplitudes computed from monthly mean TROPOMI data exhibit a pronounced seasonal cycle reflecting the varying intensity of biomass burning around the Atlantic basin (Sauvage et al., 2007). The wave-one pattern is strongest (∼41 DU) during September–November and weakest during May–June (∼26 DU). Again, OMI and GOME-2B yield similar conclusions.

Depressions in tropospheric ozone are expected above high altitude terrain. TROPOMI's mean TrOC field indeed traces

surface elevation (red isolines show 500, 1000 and 2000 masl in Fig. 1 (top)). Topographic effects are particularly clear over high mountain ranges (e.g., Andes in South America, New Guinea Highlands), but lower lying terrain (500 masl) also leaves





a noticeable imprint on mean TROPOMI TrOC field (e.g., around Gulf of Aden, equatorial West Africa). This illustrates that averaging two years of TROPOMI data reduces random TrOC error to almost negligible levels.

## 6.2 Biomass burning

Open fires of vegetation release large amounts of volatile organic compounds and nitrogen oxides (Andreae, 2019). These interact photochemically in the smoke plume and produce ozone which is transported away from the burning area (Jaffe and Wigder, 2012; Monks et al., 2015). Such biomass burning events occur primarily during the dry season in tropical regions with rainforest or savanna (Africa, South America, Indonesia) (Ziemke et al., 2009a; Leventidou et al., 2016) and affect air quality on regional scales. Monitoring the strength, spatio-temporal variability and longer-term evolution of precursor emissions and

ozone due to, e.g., biomass burning, are primary mission objectives for TROPOMI, among others to provide better constraints for data assimilation and inverse modelling (e.g., Veefkind et al., 2012; Borsdorff et al., 2020; Schneising et al., 2020; van der Velde et al., 2020; Verhoelst et al., 2020).

TROPOMI observes elevated levels of tropospheric ozone when and where expected from biomass burning, with columns of more than 35 DU and up to ∼45 DU during July–November across the Atlantic basin. Figure 12 shows median TrOC over

two week periods in 2018 and 2019. Only cells with homogeneous temporal sampling and a value above 30 DU are shown. In 2018, TrOC levels above 35 DU were recorded from early July onward over the equatorial Atlantic and these increased to more than 40 DU across the southern Atlantic around mid September until end October. The highest two-week mean column of 48 DU was located off the Angolan coast in the second half of September. Return to values below 35 DU occurred during late November. In 2019, the season was less intense and started several weeks later. TrOC values above 35 DU first appeared only

around mid September and lasted until early December, two weeks later than in 2018. Maximal values of 45 DU were observed in the first two weeks of October and the first half of November 2019 in the southern Atlantic. OMI and GOME-2B TrOC data indicate similar timing and location of elevated ozone levels (Fig. S6 in Supplement), however, the sampling resolution of TROPOMI is better than its predecessors allowing a more finely resolved monitoring.

Unusually high numbers of fires were active in Brazil during August 2019, three times more than in August 2018 and the

highest fire count of the past decade (Barlow et al., 2020). However, tropospheric ozone measurements by TROPOMI, OMI, GOME-2B and ozonesonde do not reveal a clear link with the Brazilian fires. Observed TrOC levels over South America were comparable to or lower than 2018 levels. The exception is perhaps the first half of November 2019 when satellite ozone columns above 40 DU occurred across the entire southern Atlantic basin. This contrasts somewhat with a sudden episode of low sonde readings at Natal and Ascension Island around this period (Fig. 2). More detailed analyses will be needed to verify

whether these high columns are related to the unusual fire activity in Brazil a few months earlier.

## 6.3 Seasonal cycle, Madden-Julian Oscillation and Kelvin waves

The two-year data record of TROPOMI should allow the detection of geophysical signals with periods ranging from two years down to twice the averaging window of the CCD algorithm, i.e., six days for TROPOMI (VanderPlas, 2018). Analyses of interannual variability due to El Niño Southern Oscillation (ENSO) and Quasi-Biennial Oscillation (QBO), of decadal





variability caused by the solar cycle, and of long-term trends (Gaudel et al., 2018; Ziemke et al., 2019), will become possible later in the mission or when TROPOMI data are merged with the European time series that started with GOME in 1995 (Valks et al., 2003; Heue et al., 2016). Focusing here on shorter timescales, we searched for periodic signals using the Lomb-Scargle periodogram, a Fourier-like power spectrum for irregularly sampled data (VanderPlas, 2018, and references therein). We used the fast algorithm by Press and Rybicki (1989) and tested significance at the 1% level.

Long gaps in the time series reduce spectral power in the periodogram. For CCD-derived TrOC data records, such gaps reoccur in winter in the outer tropics when insufficient numbers of convective clouds are present in the Pacific reference sector to obtain a reliable estimate of the stratospheric column (Sect. 2.2). Periodograms at Hilo (19.7° N) and Suva (18.1° S) therefore show less overall power than at lower latitudes.

   The two most powerful spectral peaks generally lie around twelve and six months. Annual and semi-annual cycles are

significant for, respectively, 88% and 75% of the TROPOMI grid cells. Both cycles are detected over about two thirds of the tropics. Spectral analysis of OMI and GOME-2B TrOC data restricted to the TROPOMI time range show significant results for similar periods and locations.

   Investigation of intraseasonal variability was limited to satellite data above the SHADOZ sites (Fig. 13). At Kuala Lumpur, four significant spectral peaks in TROPOMI data are located in the 30-60 day range, which suggests influence by the Madden-

Julian Oscillation (MJO) (Ziemke et al., 2007, 2015; Stauffer et al., 2018). An active MJO phase brings enhanced convection and therefore reduced tropospheric ozone levels. MJO activity is strongest during boreal winter months (November-February) in the warm pool of the equatorial Indian and western Pacific oceans, where the Kuala Lumpur site is located. The periodic 5-10 DU dips in TROPOMI TrOC coincide reasonably well with the occurrence of negative anomalies in outgoing longwave radiation and positive anomalies in westerly wind speeds, both indicative of a convective MJO phase (Figs. 2, bottom left

and 14)

   Inspection of TROPOMI TrOC time series (Fig. 2) suggests the presence of a high frequency signal that escapes detection by the Lomb Scargle analysis. Rapid oscillations in TrOC are particularly clear at low latitudes (Natal, Nairobi, Ascension Island), but seem present over the entire tropical belt although less frequent or less powerful. Their amplitude is about 3–5 DU in the inner tropics, their period around 1–2 weeks. The oscillation is nearly identical for TROPOMI total ozone and tropospheric

ozone column data. In contrast, a much weaker periodic behaviour is seen in the stratospheric reference columns (270 hPa to top of atmosphere) derived by the CCD algorithm (Sect. 5. The period of the TrOC signal must be changing over time since it is not picked up by the spectral analysis over the two year period. We noted similar oscillations in the GOME-2B and OMI data records as well, the sonde data are too sparsely sampled. Period, amplitude and the clear signal close to the Equator are all reminiscent of Kelvin waves (Holton and Hakim, 2013). Such waves have been observed in total ozone and stratospheric ozone

records (Ziemke and Stanford, 1994; Fujiwara et al., 1998; Timmermans et al., 2004; Feng et al., 2007). Further analysis of the TROPOMI TrOC data will be needed to ensure the oscillation resides in the troposphere, to characterise it (e.g., using wavelets or windowed Fourier transforms which avoid diffusing the quasi-periodic signal) and to falsify the Kelvin wave hypothesis.



# 7 Conclusions

Since October 2017, TROPOMI aboard Sentinel-5 Precursor is the newest member of the European family of polar-orbiting
nadir UV-Visible sounders that started with ERS-2 GOME in 1995. A suite of ozone and ozone precursor data products are
retrieved at unprecedented spatial resolution and made available to the Copernicus atmosphere monitoring and climate change
services (CAMS and C3S), to the scientific community, and to the general public. In this paper we assessed the quality of the
first two years of TROPOMI tropical tropospheric ozone columns obtained with the Convective Cloud Differential technique
(product identifier: S5P L2_O3_TCL V01.01.05–08). This data product is daily gridded at 0.5° latitude by 1° longitude
resolution between 20° S and 20° N, and represents the three-day moving mean tropospheric ozone column between surface
and 270 hPa in cloud-free conditions. The quality of the TROPOMI data record was assessed using three complementary
methods. We first compared TROPOMI data to co-located ozonesonde flights and satellite data by OMI and GOME-2B.
Small-scale patterns due to sampling errors were then inferred from detailed inspection of sequences of individual maps. And,
finally, we explored how TROPOMI perceives known geophysical signals and patterns.

All quality indicators in the comparison analysis are based on robust statistical estimators, such that outliers do not skew our
conclusions from the short and sparse comparison time series. Since the OMI and GOME-2B sounders have similar instrument
design and spectral coverage and their tropospheric ozone data are also derived using the CCD technique, these satellite records
are less independent from TROPOMI's CCD data record than the measurements by ozonesonde. TROPOMI captures the spatial
and temporal variability seen by the other instruments reasonably well. Typically, correlation with the SHADOZ ozonesonde
network is strong (61%) and with satellite data even stronger (80–90%). Correlation coefficients drop considerably (by ~10–
20%) in regions with low natural variability and smaller ozone column (e.g., western Pacific Ocean). This does not point to
a poor performance of the instrument, rather, it is the combined result of the poor dynamical range in TrOC offered by the
atmosphere and the larger relative contribution of uncorrelated random measurement errors.

Statistical dispersion in the TROPOMI comparisons is stable in time and ranges between 2.6–4.6 DU when averaged over
the tropical belt. The spread in satellite intercomparisons increases by 1 DU (and more) between equator and outer tropics,
in line with expectations from increased natural variability at higher latitudes. In fact, like in many other analyses, natural
variability within the co-location window represents an important challenge to infer random measurement uncertainty directly
from the scatter in the difference time series. We reduced its impact by using the triple co-location technique to disentangle the
measurement and representativeness components in the random error budget. Using triplets of TROPOMI, OMI and GOME-2B
columns we obtain single-measurement TROPOMI random uncertainties of 1.5 DU around the equator and 2.2–2.7 DU in the
outer tropics. Uncertainties reported in the data files exhibit a similar meridian structure and they are only slightly larger than
those inferred from triple co-locations (but not more than 0.4 DU). The reasonable agreement lends confidence that the reported
uncertainties in the data files are indeed realistic. When compared to other satellite sensors TROPOMI random uncertainties
are 0.3–0.5 DU (20–25%) smaller and signal to noise ratios are 1–2 dB larger, while offering six times better spatio-temporal
resolution. In addition, we noticed that the reported uncertainty for OMI is generally too high (by 0.3–1.0 DU), and for GOME-
2B as well in the South Atlantic Anomaly (by more than 2 DU).



Two sources of error that are correlated at small spatio-temporal scales were investigated in detail, in order to assist users desiring to exploit the TROPOMI record at maximal sampling resolution. Both types of error originate in the sampling pattern resulting from the interplay between TROPOMI's orbit, cloud coverage and geophysical variability in the ozone field. The first type is related to the sampling of deep convective cloud scenes needed to estimate the stratospheric ozone column. It leads to a banded structure in the latitude domain that persists over 1–2 weeks and is especially prominent in the outer tropics. About 10% of the measurements outside 15° latitude exhibit a TrOC error of at least 0.6–0.8 DU, while in the innermost tropics such errors occur in less than 1–2% of the record. The second type of error originates in the sampling of cloud-free scenes and results in artificial correlated patterns in tropospheric ozone close to the synoptic scale. Inspection of animated sequences of TROPOMI maps revealed that these errors appear across the entire tropical belt and that they are often oriented along and progressing with TROPOMI orbit. Although difficult to characterise, errors of more than 5 DU are seen regularly and these will clearly dominate the error budget for the affected measurements.

The high sampling resolution of TROPOMI allows to aggregate the data in time and/or space and reduce random errors to negligible levels while still preserving a resolution on par with other operational satellite sensors. Systematic error will fairly rapidly dominate the error budget for regionally or temporally averaged data. The median difference between TROPOMI and co-located reference data varies, depending on the instrument. When averaged over the network or the entire tropical belt, we find that TROPOMI has a positive bias with respect to sonde (+2.3±1.9 DU) and GOME-2B (+2.3±0.6 DU), while an insignificant negative bias is seen versus OMI (-0.1±0.7 DU). Error bars represent the statistical dispersion ($1\sigma$) of the bias estimates over the ground-based network or tropical belt. The intercomparison with OMI furthermore suggests a meridian pattern (1 DU larger TrOC in the northern tropics than in the south) and a zonal pattern (1 DU larger TrOC in the Pacific compared to the Atlantic) in TROPOMI bias. The GOME-2B results show a similar meridian dependence although weaker, but no zonal structure. The sparsity of the ozonesonde network in combination with systematic differences between stations impedes an independent confirmation of spatial patterns in TROPOMI bias at the 1 DU level.

The cause of the TROPOMI bias and its dependence on reference instrument are not fully understood at the moment and will be subject of further work. One reasonable explanation for part of the differences in bias is the systematic difference in measurement time of the instruments in the presence of a diurnal cycle. However, not much is known about the strength and character of a diurnal cycle in tropical tropospheric ozone. Diurnal variations were noticed in measurements at the surface and in the boundary layer, but not in the free troposphere. If the cycle resembles the one observed over Frankfurt then it would contribute 0.5–1 DU to the positive bias seen in GOME-2B and sonde comparisons. Biases in TROPOMI total ozone and cloud retrieval could explain part (or all) of the positive bias as well, but the former are challenging to constrain. TROPOMI bias is especially sensitive to a cloud-dependence of total ozone bias. We find hints of a 1–2.5% difference in TOC bias between very cloudy and cloud-free scenes from comparisons of TROPOMI, Brewer and SAOZ. If confirmed, this implies a $\sim$ +2–5 DU bias in TrOC. The negative bias in cloud height retrieval may impart an additional +0.5 DU. Further work is needed to corroborate these hypotheses or to attribute the bias to the reference sensor.

No gradual drift with respect to the ozonesondes, OMI nor GOME-2B was noted during TROPOMI's first two years of operation. However, the record is still fairly short and continued monitoring will be important, also because hints of two shorter-



term temporal patterns were observed. TROPOMI-satellite biases across the entire tropical belt were 1.5–2.5 DU higher during March–July 2019 than during September–January in 2018 and 2019. The pattern seems to continue in 2020 as well, with anomalies starting to increase again in the first few months. This pattern could not be confirmed from the sparse ozonesonde

intercomparisons. In addition, the three satellite records overestimated ozonesonde data around the Atlantic basin by 5–10 DU during July–November 2018. A similar high bias reappeared the next year over Paramaribo but not as clearly at the three other sites. Longer time series will be needed to clarify whether both temporal patterns in TROPOMI bias persist, whether these were episodic periods or whether these can be attributed to the reference data record or mismatch uncertainty. The latter could be caused by the first-order approximation that satellite CCD data have uniform vertical averaging kernel over the tropospheric

column, a hypothesis which is under investigation by the TROPOMI Mission Performance Centre.

Besides performing comparisons to other data records we also demonstrated the ability of TROPOMI to capture several known geophysical signals and patterns. The permanent zonal wave-one structure is clearly present in time-averaged tropical maps with a mean amplitude of 23.6 DU between the Atlantic highs and the Pacific lows. The strength of this pattern in TROPOMI data modulates seasonally, following the biomass burning season, and has maximum amplitude in September–

November and minimal values in May–June. The 2018 and 2019 biomass burning seasons are well recorded by TROPOMI, at superior sampling resolution than OMI and GOME-2B. Record high fire counts in Brazil in August 2019 do not appear to lead to record numbers in tropospheric ozone. On the contrary, 2018 ozone levels were generally higher than in 2019 across the Atlantic basin. Analysis of Lomb-Scargle periodograms unveiled significant spectral peaks for the annual and semi-annual cycles in TROPOMI data over a large part of the tropical belt. Additional peaks in the 30–60 day range were discovered

over Kuala Lumpur in the Indo-Pacific warm pool, which correspond to periodic dips of 5–10 DU in TROPOMI time series that may be attributed to enhanced convective activity associated with the Madden-Julian Oscillation. High frequency quasi-periodic signals in TrOC exhibit characteristics (1–2 week period, 3–5 DU amplitude, strongest at low latitude) reminiscent of Kelvin waves.

Our estimates of the bias (0.1–2.3 DU or 0.3–13%) and single-measurement uncertainty (<1.5–2.5 DU or ∼8–13%) demon-

strate that the studied TROPOMI tropospheric ozone column data meet the pre-launch mission requirements of <25% on the systematic error and on the precision. TROPOMI captures known patterns and variability in the tropospheric ozone field accurately, with better precision and at higher spatio-temporal resolution than its predecessors. It is therefore a particularly valuable addition to the global monitoring system, one that will allow new and more refined analyses of ozone and its precursors. With slightly longer time series and a better view on whether the temporal features unveiled in this study persist or dissolve, the

TROPOMI record has clear potential to contribute to the long-term tropospheric ozone data records required by the Global Climate Observing System (GCOS, 2016) and by the second Tropospheric Ozone Assessment Report of the International Global Atmospheric Chemistry project (TOAR-II, Gaudel et al. (2018, and references therein)). On top of its data quality and horizontal resolution, its daily coverage over the tropical belt and sampling resolution make TROPOMI well suited to serve as the travelling standard interconnecting regional tropospheric ozone measurements by the constellation of geostationary air

quality satellites (GEMS, TEMPO and Sentinel-4). To accomplish this interoperability objective and to further characterise and improve its data products, an important next step will be to investigate its mutual coherence with satellite tropospheric





ozone data inferred using different retrieval techniques (e.g., cloud slicing, and optimal estimation profiling) and also from instruments with different spectral ranges and sensitivities.

## Appendix A: Tropospheric column from sonde profile

The partial column of ozone TrOC, expressed in Dobson units, between surface and the $270\,\mathrm{hPa}$ level ($\simeq 10\,\mathrm{km}$ in the tropical belt) is obtained by integrating the screened sonde ozone volume mixing ratio profile $X_i$ over $i = \{1 \ldots N\}$ pressure levels $P_i$

$$
\begin{aligned}
\mathrm{TrOC} &= \frac{N_A k_B}{\mu_d} \frac{T_0}{P_0} \int\limits_{P_{\mathrm{surface}}}^{270\,\mathrm{hPa}} \frac{X(P)}{g(P)} dP, \\
&\simeq \frac{N_A k_B}{\mu_d} \frac{T_0}{P_0 g_0} \sum_{i=1}^{N} \frac{X_{i-1} + X_i}{2} (P_{i-1} - P_i),
\end{aligned}
\tag{A1}
$$

where $N_A$ and $k_B$ are the Avogadro and Boltzmann constants, $\mu_d$ is the molar mass of dry air, and $T_0$, $P_0$ and $g_0$ are the standard temperature, standard pressure and standard gravitational acceleration. The factor in front of the summation equals

$0.7891\,\mathrm{DU}\,\mathrm{hPa}^{-1}\,\mathrm{ppmv}^{-1}$ for $X_i$ and $P_i$ expressed in ppmv and hPa. A derivation can be found in Appendix B of Ziemke et al. (2001). Volume mixing ratio at the $270\,\mathrm{hPa}$ level is interpolated from the original sonde profile data. The partial column below the first sonde measurement is not included and assumed negligible since the first returned reading usually occurs within $100\,\mathrm{m}$ from the surface. In rare cases, the first reading is higher. If the log(pressure) range sensed by the sonde misses 3% or more of the surface–$270\,\mathrm{hPa}$ range, the sonde tropospheric ozone column is discarded.

*Data availability.* Sentinel-5 Precursor TROPOMI data are available from the Copernicus Open Access Hub at https://scihub.copernicus.eu (last access: 24 May 2020). This data set is open for use by the public, subject to the data policy. Also subject to data use policies, the ozonesonde data are publicly available from the SHADOZ Data Archive at https://tropo.gsfc.nasa.gov/shadoz (last access: 24 May 2020). Sonde data for the NOAA and KNMI stations were obtained directly from the data provider (last access: 24 May 2020). The GOME-2B and OMI data were processed by BIRA-IASB and DLR in the framework of ESA's Ozone_cci project and are available upon request (DLR).

*Author contributions.* DH conceived, coded and performed the analysis. TV analysed the TOC comparisons for CF dependence and suggested the Kelvin wave hypothesis. JCL, AK, TV and SC contributed input and discussion at all stages of the analysis. KPH, FR and DL lead the algorithm development of TROPOMI, OMI and GOME-2B tropospheric ozone data used in this work. CL lead the algorithm development of the offline total ozone retrieval. DEK, AMT, JCW, BJJ and MA maintain and provide access to ozonesonde data archives. AMT, RMS, BJJ, PDC, AP, MA, SM, HV, HS, MM, CF and FRS supervise and/or carry out ozonesonde measurements. JPV, CZ, and AD manage
the Copernicus S5p mission, the S5p MPC and/or the S5PVT. DH wrote the manuscript, JCL the introduction, and TV the TOC parts. All authors revised and commented on the manuscript.



*Competing interests.* The authors declare that they have no conflict of interest.

*Acknowledgements.* Part of the reported work was carried out in the framework of the Copernicus Sentinel-5 Precursor Mission Performance Centre (S5p MPC), contracted by the European Space Agency (ESA/ESRIN, Contract No. 4000117151/16/I-LG) and supported by the Bel-
gian Federal Science Policy Office (BELSPO), the Royal Belgian Institute for Space Aeronomy (BIRA-IASB), the Netherlands Space Office (NSO), and the German Aerospace Centre (DLR). Part of this work was also supported by the S5P Validation Team (S5PVT) AO project CHEOPS-5p (ID #28587, Co-PIs J.-C. Lambert, D. Hubert and A. Keppens, BIRA-IASB) with national funding from the BELSPO/ProDEx project TROVA-E2 (PEA 4000116692). The authors express special thanks to José Granville and Olivier Rasson for satellite- and ground-based data post-processing and for their dedication to the S5p operational validation. This work contains modified Copernicus Sentinel-5
Precursor satellite data (2018–2020) processed by DLR and post-processed by BIRA-IASB in the framework of the S5p MPC. This work also contains modified GOME-2B and OMI satellite data processed by BIRA-IASB and DLR in the framework of ESA's Ozone_cci project (http://cci.esa.int/ozone) and post-processed by BIRA-IASB. The ozonesonde data used in this publication were obtained as part of NASA's Southern Hemisphere ADditional OZonesonde programme (SHADOZ, https://tropo.gsfc.nasa.gov/shadoz) and the Network for the Detection of Atmospheric Composition Change (NDACC, https://ndacc.org), and are publicly available. The PIs and staff at the ozonesonde
stations are warmly thanked for their sustained effort on maintaining high quality measurements and for valuable scientific discussions. NOAA/NCEP/CPC is acknowledged for the provision of daily Madden-Julian Oscillation Indices.





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



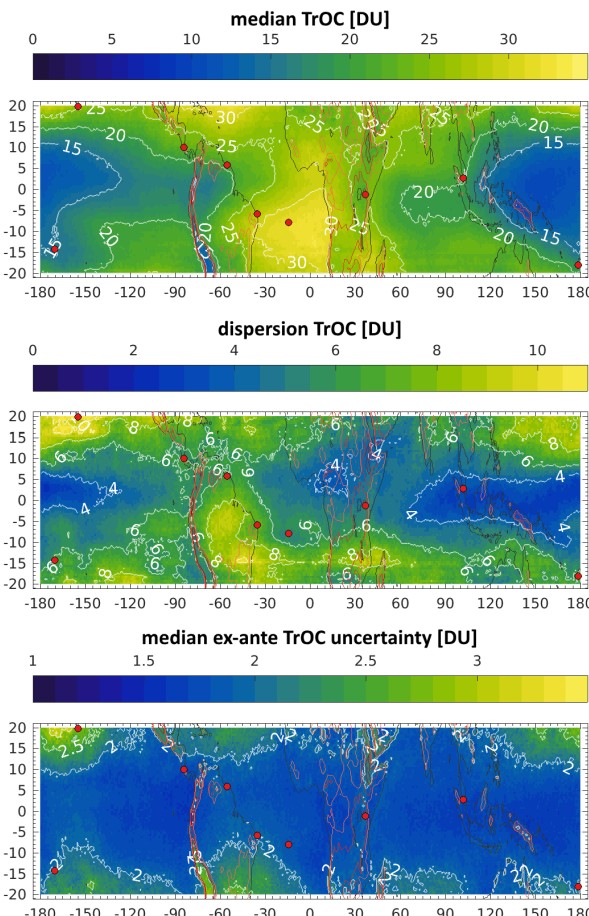

**Figure 1.** Statistics of two years of TROPOMI tropospheric O3 column data (1 May 2018 – 30 Apr 2020): (top and centre) median and 68% interpercentile of tropospheric ozone column, (bottom) median ex ante tropospheric ozone column uncertainty. Data are screened according to the recommendations given by the data providers. Red markers show the location of the SHADOZ sites considered in this study. Red isolines trace the 500, 1000 and 2000 m surface elevation levels. Corresponding sampling statistics and time are shown in Fig. S1 in Supplement.



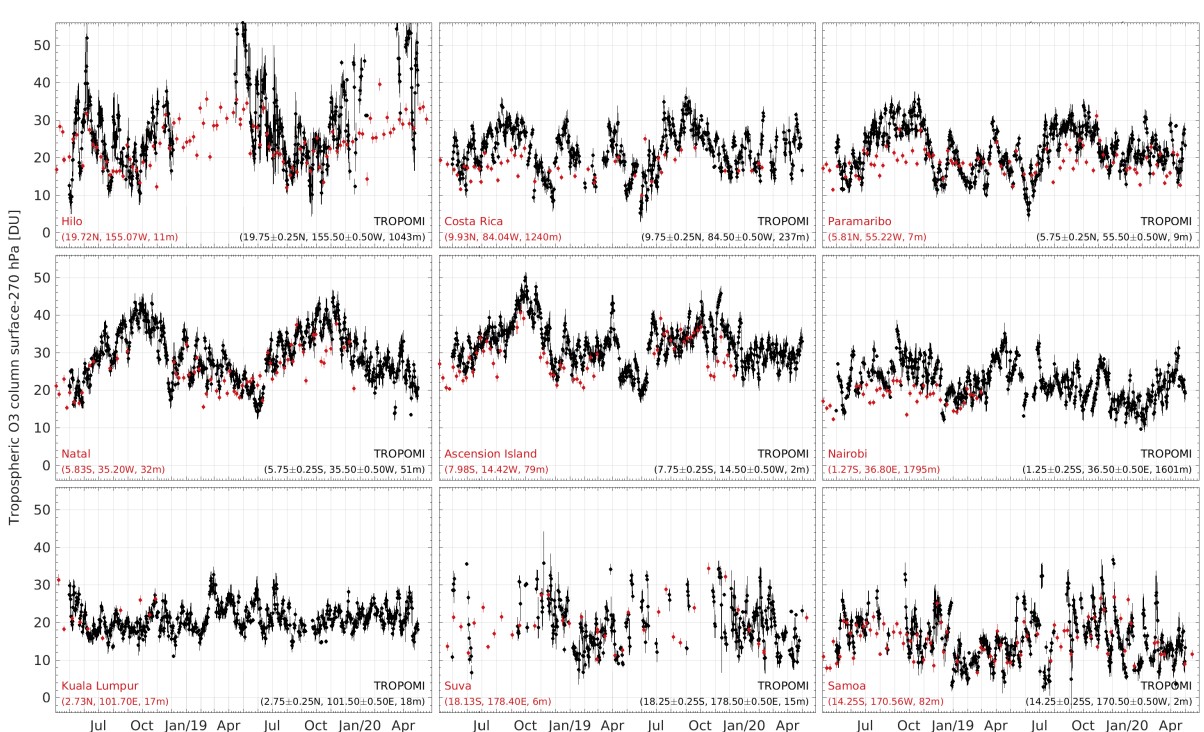

**Figure 2.** Time series of spatially co-located TROPOMI (black) and ozonesonde (red) tropospheric ozone column data over nine SHADOZ sites.

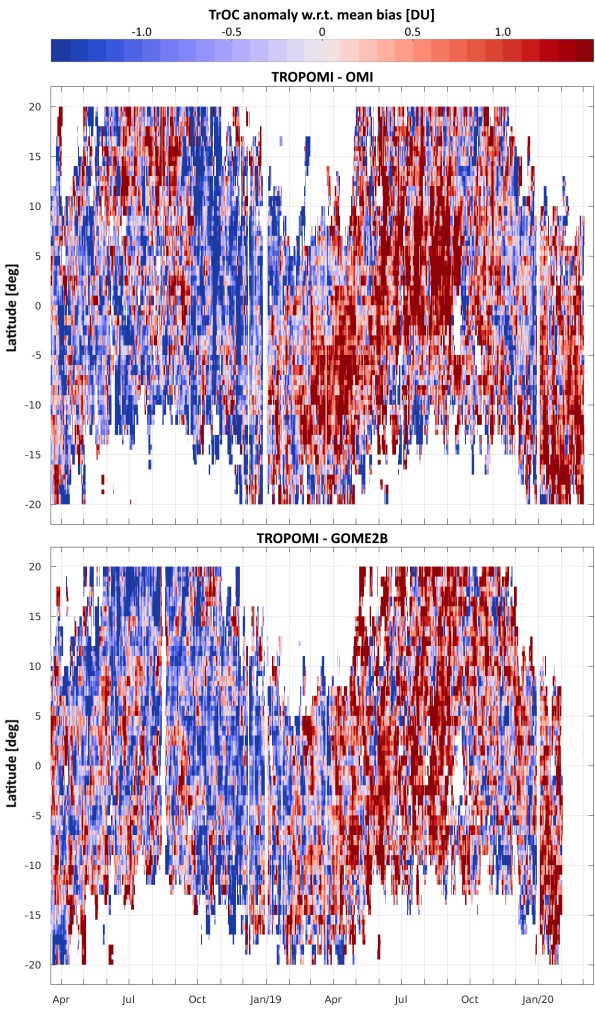

**Figure 3.** Latitude/time section of the anomaly of the TROPOMI tropospheric ozone column with respect to its mean bias with OMI (top) and GOME-2B (bottom) from April 2018 through February 2020. Zonal mean differences of the bias have been subtracted in order to better highlight the meridian coherence of the change over time.



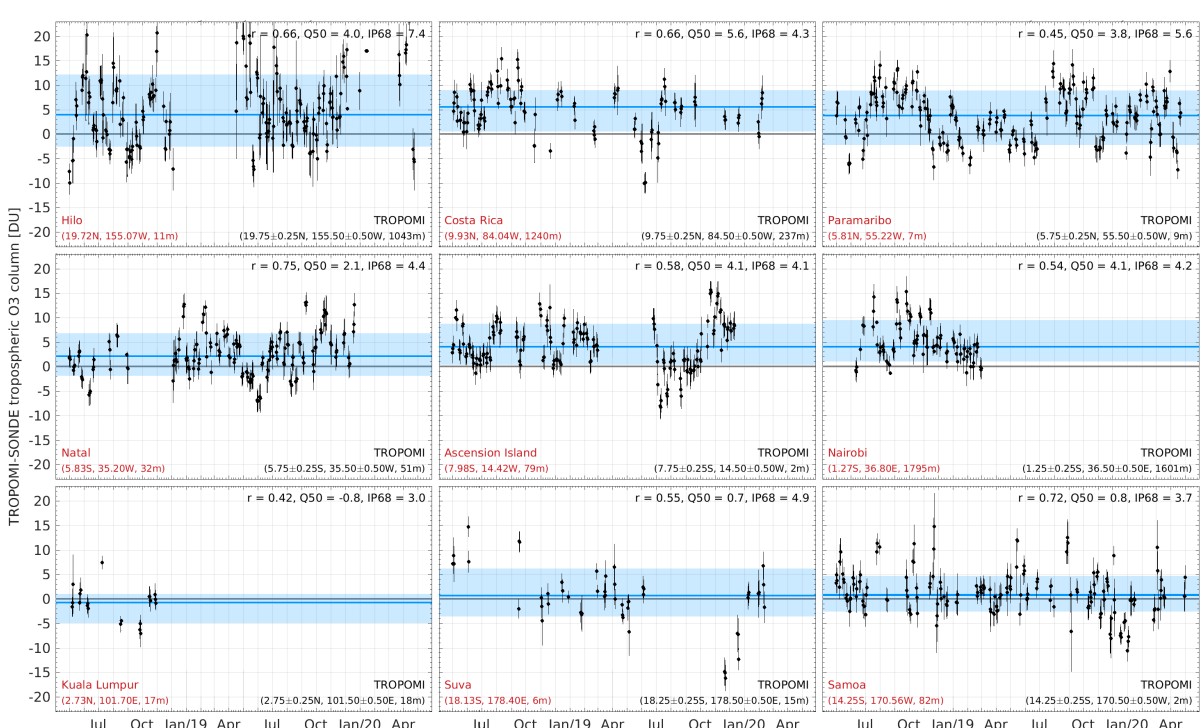

**Figure 4.** Time series of the difference between spatially and temporally co-located TROPOMI and ozonesonde tropospheric ozone column data over nine SHADOZ sites. Positive values indicate that TROPOMI overestimates the ozonesonde value. Blue line and area show median and 68% interpercentile over the entire period.

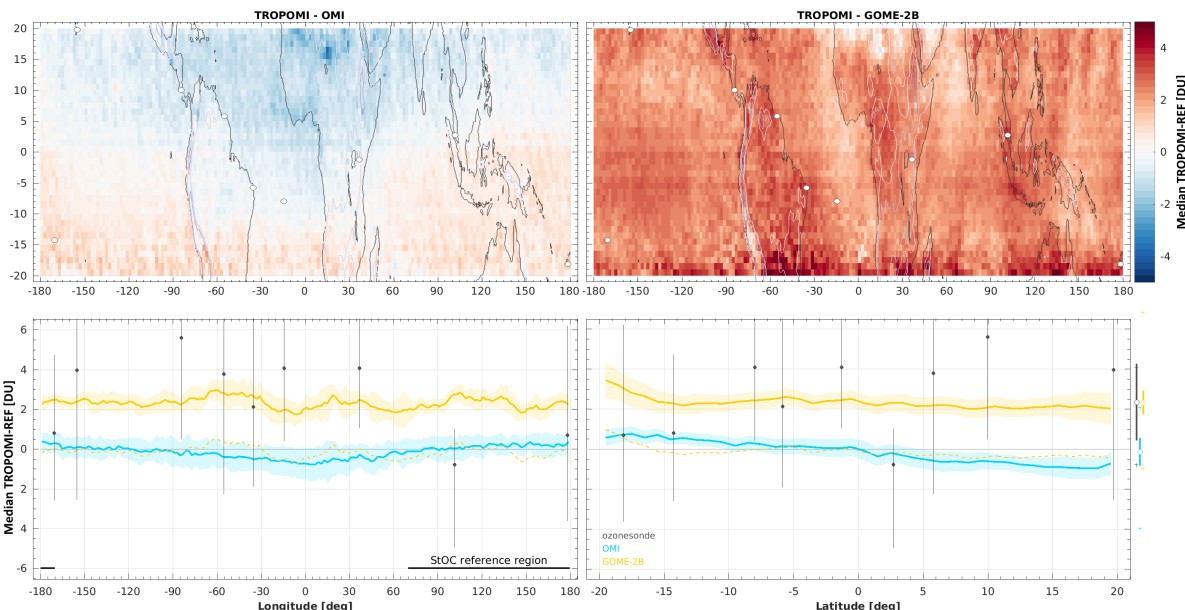

**Figure 5.** Top row: Median difference between spatially and temporally co-located TROPOMI tropospheric ozone column and OMI (left) or GOME-2B (right) data; contours trace isolines of surface elevation. Bottom row: longitude (left) and latitude (right) section of mean and standard deviation (1σ) of the ΔTrOC map. Black markers display median and 68% interpercentile of co-located TROPOMI minus SHADOZ ozonesonde data. Positive values indicate that TROPOMI TrOC are biased high with respect to the reference data. GOME-2B results are also offset (yellow dashed) to those found with OMI to facilitate comparison of the spatial structure. Mean and standard deviation (1σ) of the TROPOMI bias estimates over the ground-based network (sonde) or tropical belt (satellite) are displayed outside the axis on the right, together with minimum and maximum values (plus signs).





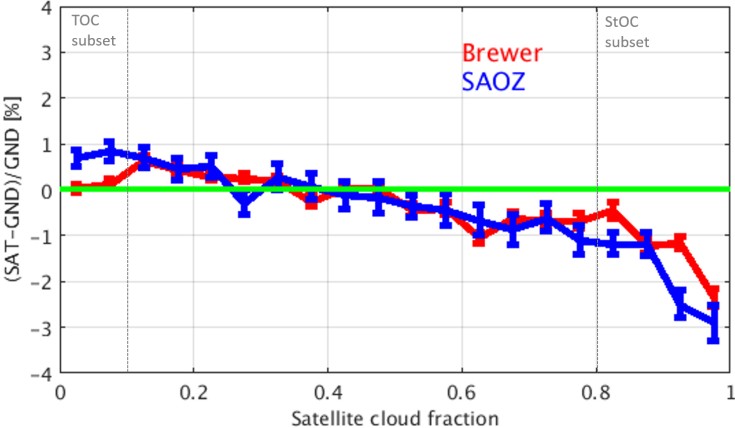

**Figure 6.** Dependence of TROPOMI total ozone column bias on the cloud fraction over the observed scene (TROPOMI L2_O3 OFFL V01.01.01-01.01.08). Comparisons are done over a two year period with respect to co-located total ozone column data from the Brewer and SAOZ monitoring networks. The two curves are normalised (on a per-station basis) on the mean difference over the 0.2–0.6 CF range, in order to reduce the station-to-station scatter of the mean bias. The two dashed vertical lines identify the cloud-free and cloudy subsets used by the CCD algorithm.

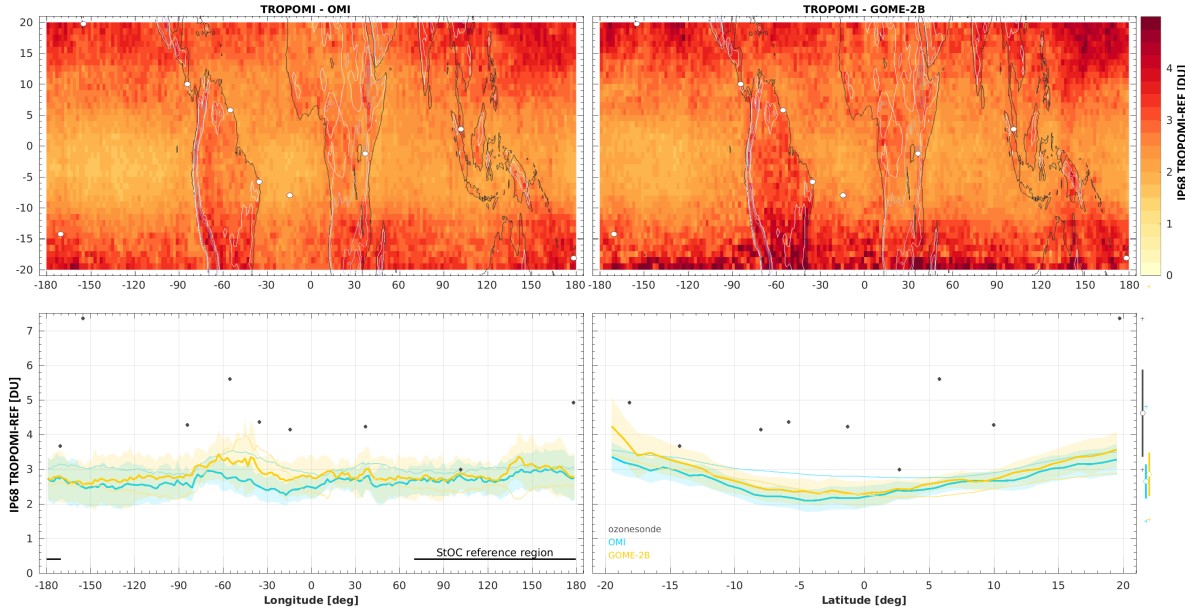

**Figure 7.** Like Fig. 5, but for the 68% interpercentile of the differences (dispersion). Bottom: Thin lines show combined ex ante uncertainty (1σ) for the satellite intercomparison. Bottom right: Mean and standard deviation (1σ) of the dispersion estimates over the ground-based network (sonde) or tropical belt (satellite) are displayed outside the axis, together with minimum and maximum values (plus signs).



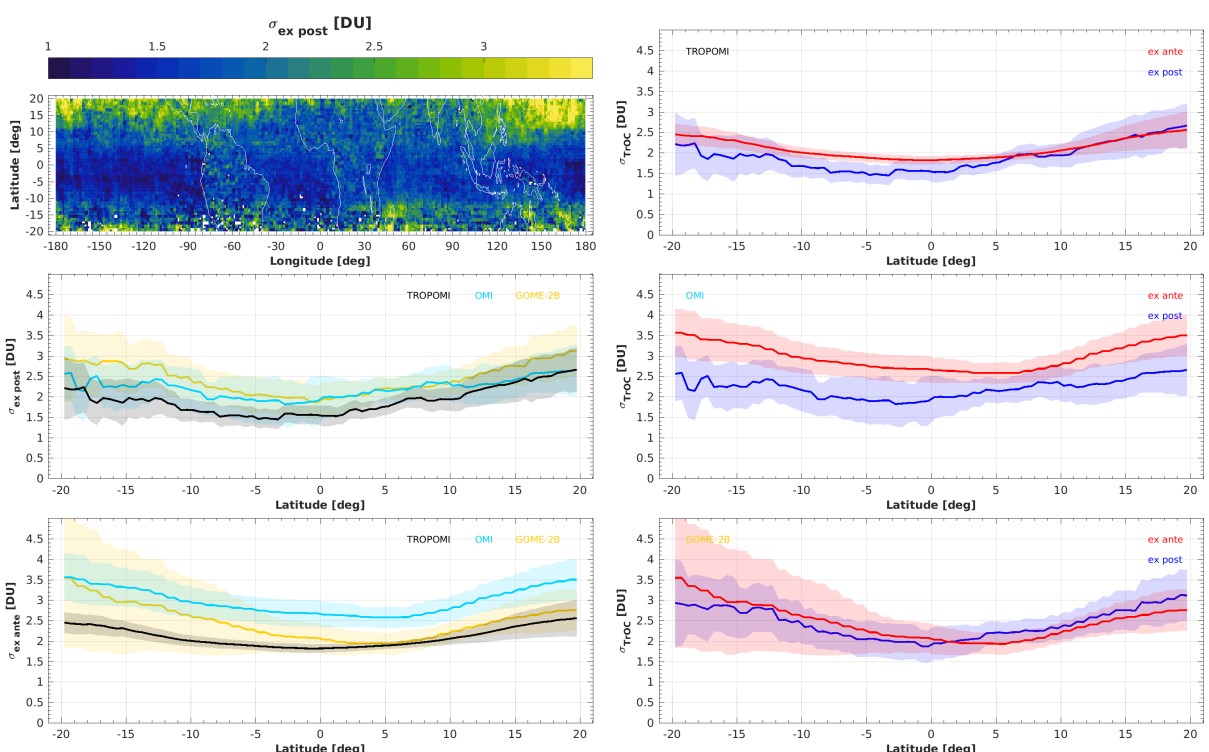

**Figure 8.** Top left: TROPOMI random measurement uncertainty estimated from co-located TrOC triplets TROPOMI, OMI, GOME-2B. Other panels: Latitudinal section of estimated (ex post) and reported (ex ante) random measurement uncertainty for the three sensors. Shaded areas indicate one standard deviation over the zonal domain.





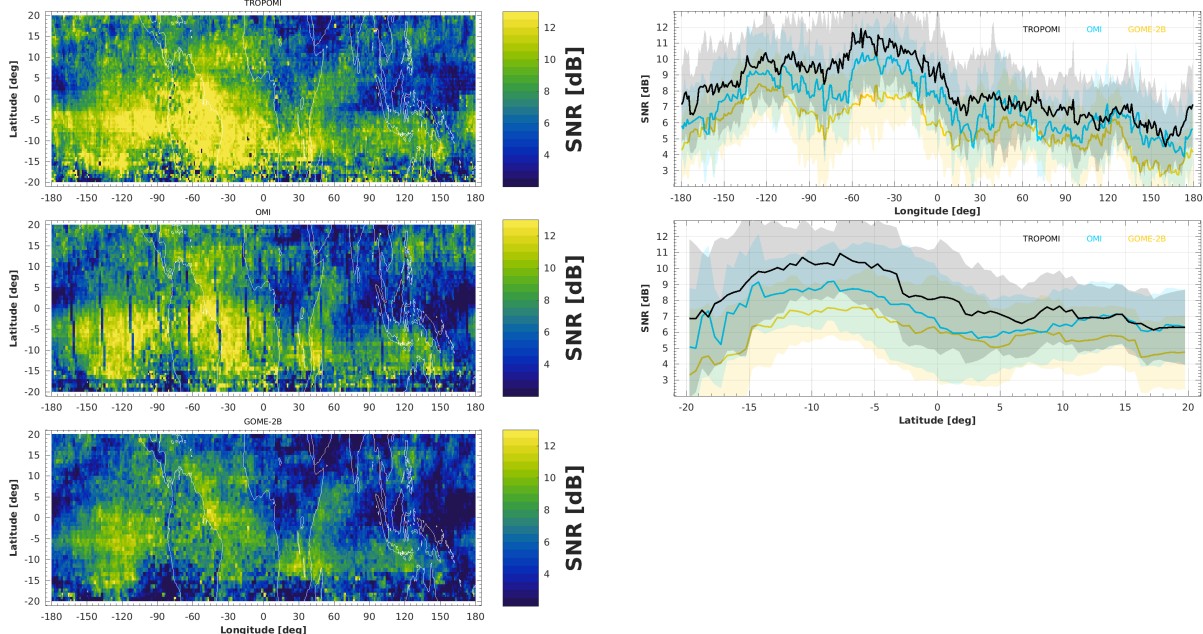

**Figure 9.** Maps (left) and latitudinal sections (right) of signal-to-noise-ratio for TROPOMI (top), OMI (centre) and GOME-2B (bottom). Shaded areas indicate one standard deviation over the zonal domain.

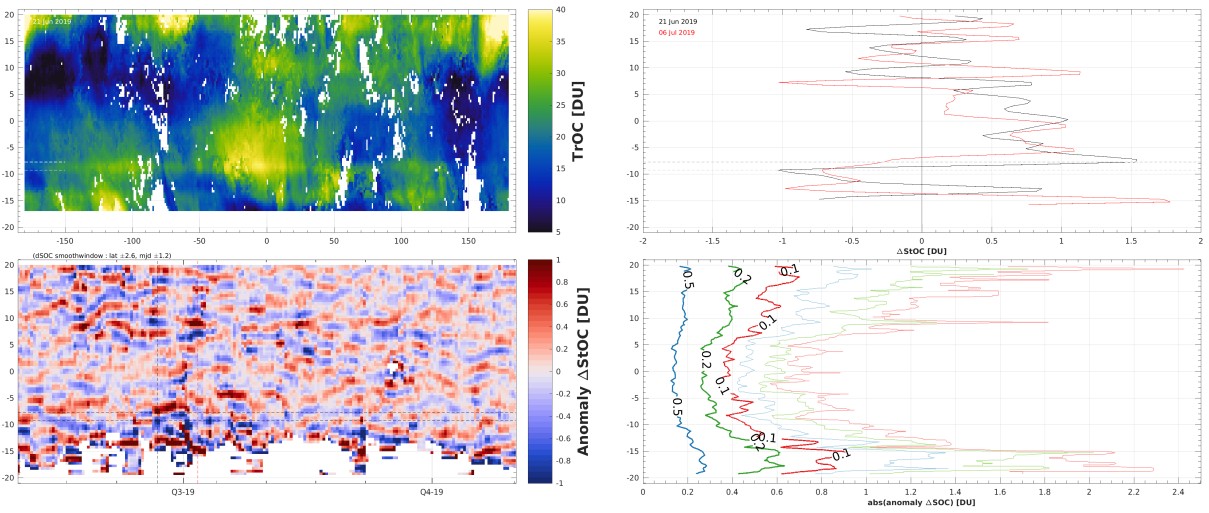

**Figure 10.** Top left: Example of banded structure around $10°$ S in the TROPOMI TrOC map of 21 June 2019. Top right: Meridian gradient of StOC for two days ($\Delta$StOC). Bottom left: Anomaly of $\Delta$StOC w.r.t. latitude- and time-smoothed $\Delta$StOC (May-October 2019). Bottom right: Cumulative distribution function of absolute value of $\Delta$StOC anomaly (thin lines correspond to 5%, 2% and 1% quantile).





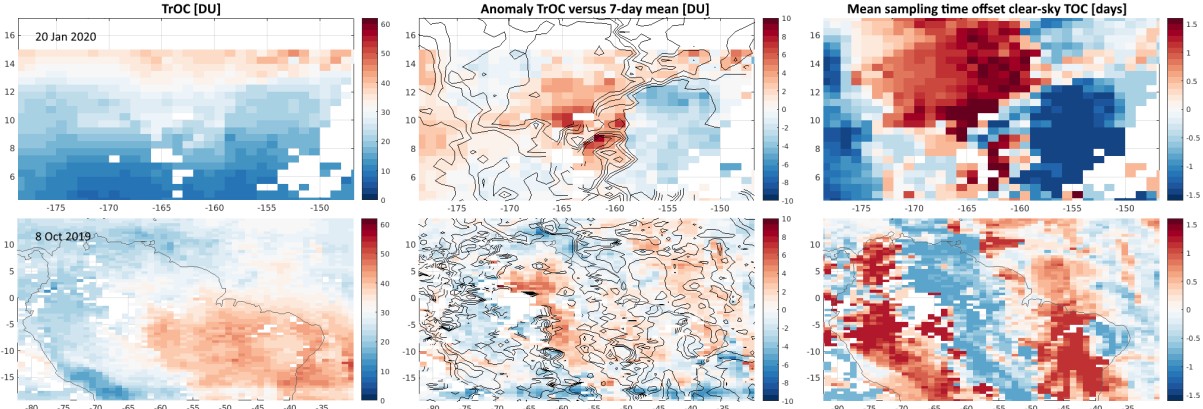

**Figure 11.** Illustration of sampling uncertainty in a region of the TROPOMI TrOC map (left) of 20 January 2020 (top row) and 18 October 2019 (bottom row). Centre: absolute TrOC anomaly relative to a seven-day moving mean; contours trace isolines of sampling time offset. Right: mean sampling time offset relative to the centre of the averaging window for the clear-sky total ozone columns used by the CCD algorithm. The structure in the mean sampling time field agrees well with that of the TrOC anomaly.

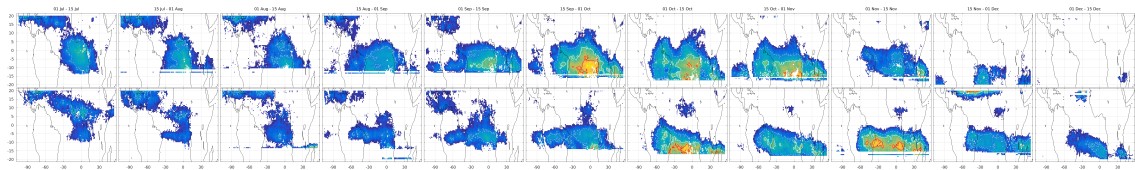

**Figure 12.** Median biweekly TROPOMI TrOC over Atlantic basin between early July and mid December (left to right) for 2018 (top rows) and 2019 (bottom rows). Grid cells with sparse or inhomogeneous temporal sampling or a value below 30 DU are blanked. Contours indicate the 35 DU (dashed), 40 DU (solid) and 45 DU (red) isolines.



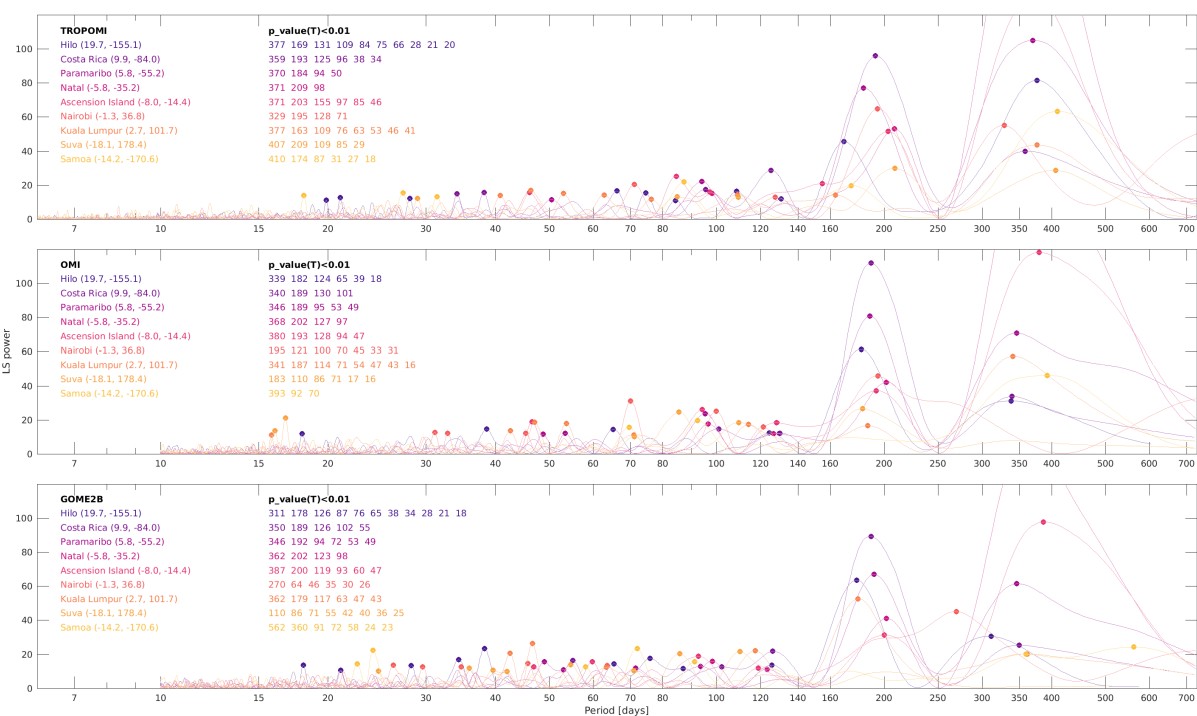

**Figure 13.** Lomb-Scargle periodogram for two years of tropospheric ozone columns over nine SHADOZ sites (coloured) for TROPOMI (top), OMI (centre) and GOME-2B (bottom). Markers locate spectral peaks that cross the 1% significance threshold (red line). Annual and semi-annual cycles appear clearly over most of the sites.

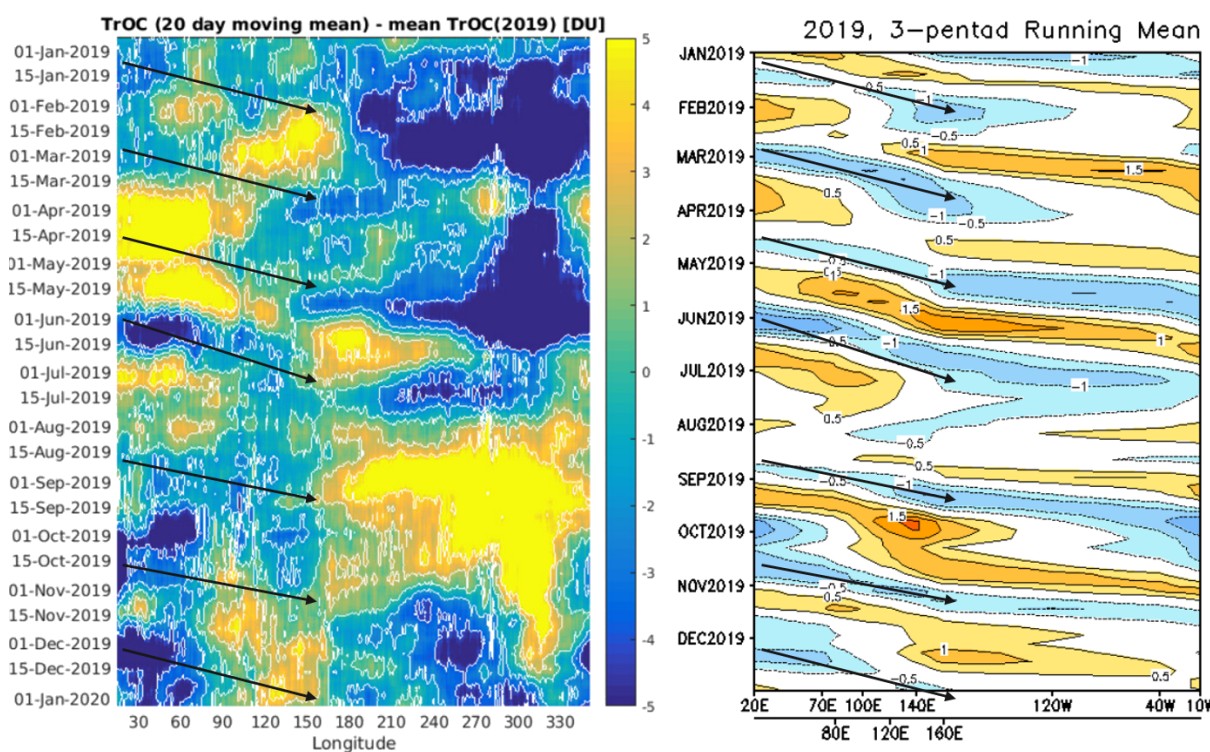

**Figure 14.** Anomaly of the 20-day moving mean TROPOMI tropospheric ozone column (left) and the NOAA/CPC Madden-Julian Oscillation index (right) during 2019 over the equatorial Indo-Pacific Oceans. Arrows indicate periodical, eastward propagating enhancements in convective activity due to MJO and the corresponding low levels of tropospheric ozone.