# Peer review of "TROPOMI tropospheric ozone column data: Geophysical assessment and comparison to ozonesondes, GOME-2B and OMI"

_Atmospheric Measurement Techniques, 2020_

## Referee Comment (RC1) · Anonymous Referee #1 · 24 Aug 2020

This paper presents the evaluation of TROPOMI tropical tropospheric ozone columns by comparison with ground based (SHADOZ) and other satellite (OMI and GOME2) data. The subject is suitable for publication in AMT and the results are of interest for the users of these data. The paper is well organised and provides valuable information about TROPOMI O3 data. The analysis of error sources is interesting. It clearly highlights the limitations of the CCD retrieval method for UV sensors based on strong assumptions about the variability of the stratospheric column of O3 and on the deep convective cloud cover. I recommend this paper after the issues listed below are dealt with.

[Figure]

General comments:

The manuscript is a bit long and could be shortened to improve its readability and efficiency. For instance, section 4.1.1 about the validation of data is too general and too long and does not provide practical and useful information and figures. Section 6 concerning "geophysical information" is providing useful results (e.g. Wave-One, Biomass Burning) but also contains less convincing information about the MJO and Kelvin waves impacts on tropospheric O3 that should be improved or removed (see details below). The conclusion (3 pages) is rather long and detailed. It should be shortened keeping the most prominent and strong results and skipping more hypothetical ones as much as possible. The authors state that "TROPOMI . . . retrievals have reduced sensitivity close to the surface and increased sensitivity above clouds". There is therefore probably a large sensitivity gradient within the troposphere itself with high sensitivity in the upper troposphere and almost none in the boundary layer. Such gradient is probably a large source of error and responsible for large discrepancies between sonde and retrieved data. But the smoothing error is not evaluated and its impact is not taken into account for the sonde versus TROPOMI comparisons. I find it ashamed because a large effort is made to quantify the other sources of uncertainty. It could interesting that the authors provide some more information about this issue and discuss the possible impact of the smoothing error on their results.

Detailed comments:

- p2l38:"some O3 is released by soils and plants". Do the authors mean that O3 is a primary pollutant directly produced by soils and plants? Could you provide references?

- section 4.4:TROPOMI is compared to OMI and GOME-2 and the biases are in good agreement with the OMI versus GOME-2 biases reported by Heu et al. (2016). The authors also document different spatial structure of the TROPOMI versus OMI and GOME-2 biases. Are there some previous studies reporting the spatial structure of the GOME-2 versus OMI bias which would support this result? Do they show a better

Interactive
comment

agreement between sondes and GOME-2 than between sondes and OMI as suggested by the similar biases from TROPOMI with sondes and GOME2?

-p10L296:"assuming that the diurnal cycle in the tropics ressemble the one over Frankfurt". Is this assumption important? Could you justify it with some references?

-p11l314:"assuming these hold for tropical conditions". Could you justify this strong assumption?

-p12l1:"r ranges 45- 75%" with SHADOZ sonde data. This values seem rather weak compared to validation studies of other satellite sensors. Could you compare your results with other trospephric ozone validation studies for e.g. OMI/GOME-2/IASI and TES in the tropical band (low O3 variability) to put them into some perspective?

- p16l491:could you explain what is a "non-geophysical" outlier and how you identify it?

-p21l644 and Fig. 13: "at Kuala Lumpur four significant peaks... in the 30- 60 days". From Fig. 13 we see some significant peaks in the 30- 60 days at other locations as well. This does not corroborate th MJO impact at KL.

- p21l648 and Fig. 14:"the periodic ... dips in TROPOMI TrOC coincide reasonably well...(Fig. 2, bottom left and 14)". Looking at Fig. 14 the coincidence between TrOC dips and the MJO Index is not very clear. On the contrary,the first and second arrows of deep convection eastward propagation do not coincide with really depleted TrOC and the third one starting on 15 April 2019 even coincide with enhanced TrOC! Furthermore, the large region of enhanced TrOc extending from 180 to 330E from 15 August to 1st November is not related to the MJO Index. The authors should strenghten their analysis of MJO-TroC relationship.

- p21l651 to l662:this part concerning Kelvin waves is not convincing at all. The relation between TrOC high frequency variability and Kelvin waves is only supported by the statement "Period, amplitude... are all reminiscent of Kelvin waves". These are rather light arguments! The authors suggest "Further analysis...nedded". I suggest to remove

this part and to keep it for another publication.

---

## Referee Comment (RC2) · Anonymous Referee #2 · 14 Sep 2020

This is an excellent paper, and should be published, essentially "as is". I have a few minor suggestions, listed below, that the authors may wish to consider.

Minor comments:

Line 62: "...systematic error and an uncertainty..."? Do you mean "systematic and random uncertainty? An error is usually something you can measure, and correct.

Line 93: "Rather THEY ARE..." (Data are plural).

Line 117: "...since THEY ARE not publicly released"

Line 137: "of systematic error in TrOC lies in a SYSTEMATIC DIFFERENCE BE-

TWEEN clear-sky and fully cloudy TOC. While IT IS challenging..."

Line 158: The terms "ex ante" and "ex post" will not be familiar to most readers and the authors might like to define them.

Line 164: "AN effective vertical resolution..."

Lines 184-185: "In reality...". This sounds as though the authors feel that the homogenization process is oversold. It would be more positive to say something like "This has reduced residual systematic differences to about 5%...".

Line 239: You might wish to comment on the variability you expect to see within this three-day window, or at least refer to section 4.1.3.

Line 273: "In general, the estimation of measurement uncertainty is more intricate than the measurement itself." An odd statement, and I'm not sure how you would justify it.

Line 275: "It is therefore good practice not to use these uncertainty estimates blindly." A bit patronizing... unfortunately the blind don't know what they can't see.

Line 285: Delete "component in the".

Line 300: "to THE TROPOMI overpass time".

Lines 302-309: Can you give a rough estimate of the range of magnitude of this error? Is it important?

Line 319: "THE smallest random uncertainty..." Line 320: Is there any other type of footprint than on the ground?

Line 352: "...cyclic pattern, OR whether it was..."

Line 384-387: "More likely" seems a strong statement. While it is possible, it is a speculation, and the Stauffer paper deals with stratospheric differences.

Lines 414-429: This seems like an important result that should be noted in the abstract.
Line 448: Not sure what a "meridian dependence" is. Do you mean "latitudinal dependence"?

Line 464: "...none of these should have superior calibration..." Is this a condition or assumption for the validity of the method?

Lines 222, 224 555: "between THE surface".

Line 55: "cloud-free".

Line 587: "differ"

Line 586: "THE location... two-year..."

Line 589: "...lack of ozonesonde stations..."

Lines 658, 682: Semicolon, not comma.

Lines 670-671: This is different from the vertical integrations used by GOME and OMI, which integrate from the surface to 10 km. I didn't see any discussion of these differences. Did I miss something?

Figure 4: I think the caption should define Q50 and IP68. "Blue line and area show median (Q50) and 68% interpercentile (IP68) over the entire period."

Figure 5: The yellow dashed line is hard to see.

Figure 14: What does "3-pentad running mean" describe exactly? I presume it's 15 of something... but it mostly tells me that the authors wish to display their knowledge of esoteric English!

---

## Author Comment (AC1) · 12 Jul 2021

**Author response to review by Anonymous Referee #1 of**

**"TROPOMI tropospheric ozone column data: Geophysical assessment and comparison to ozonesondes, GOME-2B and OMI"**

D. Hubert *et al.* ([daan.hubert@aeronomie.be](mailto:daan.hubert@aeronomie.be))

This paper presents the evaluation of TROPOMI tropical tropospheric ozone columns by comparison with ground based (SHADOZ) and other satellite (OMI and GOME2) data. The subject is suitable for publication in AMT and the results are of interest for the users of these data. The paper is well organised and provides valuable information about TROPOMI O3 data. The analysis of error sources is interesting. It clearly highlights the limitations of the CCD retrieval method for UV sensors based on strong assumptions about the variability of the stratospheric column of O3 and on the deep convective cloud cover. I recommend this paper after the issues listed below are dealt with.

**General comments**

The manuscript is a bit long and could be shortened to improve its readability and efficiency. For instance, section 4.1.1 about the validation of data is too general and too long and does not provide practical and useful information and figures. Section 6 concerning "geophysical information" is providing useful results (e.g. Wave-One, Biomass Burning) but also contains less convincing information about the MJO and Kelvin waves impacts on tropospheric O3 that should be improved or removed (see details below). The conclusion (3 pages) is rather long and detailed. It should be shortened keeping the most prominent and strong results and skipping more hypothetical ones as much as possible.

> We appreciate Reviewer #1's concern that the manuscript is somewhat on the long side. Reviewer #2, on the other hand, appears quite happy with the manuscript and had no remarks related to the readability. We consulted our team of 25+ co-authors and, generally, the readability & efficiency of the manuscript was positively evaluated. We would therefore like to leave it to the editor's judgement whether the balance between conciseness and detail is optimal for the AMT audience, or not.
>
> In the meantime, we removed the paragraphs and statements related to Kelvin waves from the manuscript (see also our responses to related questions below), and improved the discussion of the link between MJO and tropospheric O3.

The authors state that "TROPOMI ... retrievals have reduced sensitivity close to the surface and increased sensitivity above clouds". There is therefore probably a large sensitivity gradient within the troposphere itself with high sensitivity in the upper troposphere and almost none in the boundary layer. Such gradient is probably a large source of error and responsible for large discrepancies between sonde and retrieved data. But the smoothing error is not evaluated and its impact is not taken into account for the sonde versus TROPOMI comparisons. I find it ashamed because a large effort is made to quantify the other sources of uncertainty. It could be interesting that the authors provide some more information about this issue and discuss the possible impact of the smoothing error on their results.

Quantification of the uncertainty in the comparison of TROPOMI and ozonesonde due to the vertical sensitivity of TROPOMI is a challenging task. This challenge is clarified in the revised manuscript. In essence, this is because the CCD algorithm aggregates a large (~100-1000) number of single measurements with different vertical smoothing properties. The CCD technique is very different from an optimal estimation retrieval for which methods to compute an averaging kernel and to estimate the vertical smoothing error in comparisons are well established (e.g., Keppens *et al.*, 2019). For CCD data, a dedicated analysis is required to propagate the error through the measurement system in a proper way. This falls outside the scope of this work, but it is currently being considered by the TROPOMI Mission Performance Team. At this point, it is too premature to give an estimate of magnitude, its sign or even the nature (systematic, random) of the smoothing error.

The figure below shows the vertical averaging kernel (left column) and apriori (right column) for TROPOMI total ozone columns by the OFFL retrieval algorithm in the tropical belt for orbit #18391. Retrievals over scenes with convective clouds (top row) or clear-sky scenes (bottom rom) display very different vertical smoothing properties. In cloudy pixels, the total ozone retrieval is 20% oversensitive just above the cloud top, which inherently leads to a location- and time-dependent shape of the averaging kernel. In clear-sky pixels, less than 50% of the information below 700 hPa (~3 km) comes from the measurement. The rest is derived from a tailored ozone climatology that uses the vertical distribution of the Aura MLS-sonde profile climatology by Labow *et al.* (2015) but normalises the tropospheric levels to the Aura MLS-OMI tropospheric column climatology by Ziemke *et al*. (2011).

[Figure]

**Detailed comments**

p2l38: "some O3 is released by soils and plants". Do the authors mean that O3 is a primary pollutant directly produced by soils and plants? Could you provide references?

> Our statement was incorrect and is now removed from the manuscript. Ozone is *deposited* on soils and plants (e.g., review by Monks *et al.*, 2015).

section 4.4: TROPOMI is compared to OMI and GOME-2 and the biases are in good agreement with the OMI versus GOME-2 biases reported by Heue *et al.* (2016). The authors also document different spatial structure of the TROPOMI versus OMI and GOME-2 biases. Are there some previous studies reporting the spatial structure of the GOME-2 versus OMI bias which would support this result? Do they show a better agreement between sondes and GOME-2 than between sondes and OMI as suggested by the similar biases from TROPOMI with sondes and GOME2?

> We are not aware of any published bias results for GOME-2B tropospheric ozone data apart from Heue *et al.* (2016), nor of any peer-reviewed reports of the spatial structure of the bias between GOME-2B and OMI TrOC data.
>
> Heue *et al.* (2016) do not elaborate on the agreement of GOME-2B or OMI with ozonesonde data. Instead, they report biases between individual satellite sensors: GOME-2B versus SCIAMACHY TrOC bias is -2.2 DU; OMI versus SCIAMACHY bias is +0.8 DU (Table 1). Hence, Heue *et al.* observed a positive bias of OMI versus GOME-2B of about 3 DU. Our estimate of a +2 DU offset between these two sensors is fairly similar. It is outside the scope of this work to clarify why our bias results and those of Heue *et al.* (2016) differ. It may be related to the use of different total ozone retrieval algorithms (GODFIT V3 or V4), different time periods or different sampling resolutions, …

p10L296: "assuming that the diurnal cycle in the tropics resemble the one over Frankfurt". Is this assumption important? Could you justify it with some references?

> The phrases leading to this statement argue that there are no published studies of the diurnal cycle in free tropospheric ozone over the tropical belt. What systematic error may be expected is therefore estimated from the only observation-based diurnal study that we could find in the literature, over Frankfurt. The importance of our statement can therefore not be evaluated.
>
> We rephrased to "If the diurnal cycle in the tropics resembles the one over Frankfurt…".

p11l314: "assuming these hold for tropical conditions". Could you justify this strong assumption?

> We could not find any observation-based study of the correlation time or length scale of free tropospheric ozone in the tropics. For this reason, we based ourselves on the findings by Liu *et al.* (2009) over Europe/US. Our assumption would even be that correlation scales in the troposphere are larger both in time and in space for the tropics. The wave-one is the dominant feature, it is persistent, and there are minimal stratospheric intrusions to disrupt the nominal structure of tropospheric ozone in the tropics.
>
> We deliberately chose to write that "[Assuming these hold for …], we expect that…" to avoid overstating that "[the] temporal smoothing difference is the main contributor to random representativeness error". We rephrased to "If these [correlation scales] hold for tropical conditions…".

p12l1: "r ranges 45-75%" with SHADOZ sonde data. These values seem rather weak compared to validation studies of other satellite sensors. Could you compare your results with other tropospheric ozone validation studies for e.g. OMI/GOME-2/IASI and TES in the tropical band (low O3 variability) to put them into some perspective?

We extended the discussion by quoting correlation coefficients found in our own analysis of GOME-2B and OMI versus ozonesonde (in Supplement) and in a validation report of the Ozone_cci project. These estimates are in line with each other. In addition, we mention results by the IASI team of 80-90% correlation between IASI and ozonesonde or FTIR (Boynard *et al.*, 2018; Fig. 14). These higher correlations are likely a result of the comparison of more tightly co-located single measurements, which introduces less noise in the comparison.

p16l491: could you explain what is a "non-geophysical" outlier and how you identify it?

The term "non-geophysical" was removed as it causes unnecessary confusion, indeed. The manuscript describes that we use the Hampel identifier, which is a robust alternative to a four-sigma outlier detection method.

p21l644 and Fig. 13: "at Kuala Lumpur four significant peaks... in the 30- 60 days". From Fig. 13 we see some significant peaks in the 30-60 days at other locations as well. This does not corroborate the MJO impact at KL.

This Comment relates to the next one. Our response can be found below.

p21l648 and Fig. 14: "the periodic ... dips in TROPOMI TrOC coincide reasonably well...(Fig. 2, bottom left and 14)". Looking at Fig. 14 the coincidence between TrOC dips and the MJO Index is not very clear. On the contrary, the first and second arrows of deep convection eastward propagation do not coincide with really depleted TrOC and the third one starting on 15 April 2019 even coincide with enhanced TrOC! Furthermore, the large region of enhanced TrOC extending from 180 to 330E from 15 August to 1st November is not related to the MJO Index. The authors should strengthen their analysis of MJO-TrOC relationship.

We performed further analysis into linking the intra-seasonal variability in TROPOMI TrOC data to Madden Julian Oscillation. A more elaborate discussion is included in the revised manuscript. We now consider three complementary MJO indices since conclusions related to timing, strength and even the presence of an MJO event may differ depending on the index used (e.g., Kiladis *et al.*, 2014).

All three indices result from an EOF analysis of dynamical or convective proxies : (1) the NOAA CPC index based on 200 hPa velocity potential, (2) the OMI[1] index based on outgoing longwave radiation (Kiladis *et al.*, 2014), and (3) the RMM index based on outgoing longwave radiation, 850 hPa zonal wind and 200 hPa zonal wind (Wheeler & Hendon, 2004). The latter two indices are usually displayed as amplitude-phase diagrams, where amplitude scales with convective activity and phase is (loosely) linked to longitude.
* * *
[1] A bit unfortunate, but OMI, here, stands for *OLR-based MJO index*. Elsewhere in the manuscript, it refers to *Ozone Monitoring Instrument*.

The top panel in the graph below shows a Hovmöller diagram of 15-day moving mean TROPOMI TrOC anomaly averaged over the inner tropics (5° S – 5° N). Superimposed are contours of the NOAA CPC index (arb. units) between -0.5 (white) and -2 (black) in steps of 0.5, low index values indicate high levels of convective activity. The bottom panels display TrOC anomaly (black) and CPC index (purple) time series at four longitudes across the equatorial Indian and Pacific oceans. Gaps in the RMM (orange) and OMI (yellow) indices occur when the considered longitude is not located in their phase sector. For sake of visibility, all MJO indices are scaled by a factor of two and their sign is such that strong convective activity corresponds to more negative displayed values.

Two strong events with a build-up followed by a depletion of tropospheric ozone are noted over the Indo-Pacific warm pool. TROPOMI TrOC increases then reduces by about 5 DU in March 2019 between 100° E – 180° E and by more than 8 DU in June 2019 over the Indian Ocean (40° E - 120° E). In between these events, in May, an excursion of ~5 DU is noted over a smaller area (80° E - 110° E). During each of these periods the CPC, RMM and OMI indices indicate suppressed convective activity during the TrOC build-up phase and strong convective activity during the depletion phase, thereby linking these changes in TROPOMI TrOC levels to MJO. At other times, depleted TrOC levels are noted as well but these are of smaller magnitude and over a smaller area. Further analysis is needed to find out whether these events are related to MJO.

[Figure]

p21l651 to l662: this part concerning Kelvin waves is not convincing at all. The relation between TrOC high frequency variability and Kelvin waves is only supported by the statement "Period, amplitude… are all reminiscent of Kelvin waves". These are rather light arguments! The authors suggest "Further analysis... needed". I suggest to remove this part and to keep it for another publication.

It was not our original intention to overstate the hypothesis that Kelvin wave activity is observed in the TROPOMI TrOC data. Additional, more detailed studies are indeed required. We removed this paragraph and all statements corresponding to Kelvin waves from the manuscript.

---

## Author Comment (AC2) · 12 Jul 2021

**Author response to review by Anonymous Referee #2 of**

**"TROPOMI tropospheric ozone column data: Geophysical assessment and comparison to ozonesondes, GOME-2B and OMI"**

D. Hubert *et al.* ([daan.hubert@aeronomie.be](mailto:daan.hubert@aeronomie.be))

This paper presents the evaluation of TROPOMI tropical tropospheric ozone columns by comparison with ground based (SHADOZ) and other satellite (OMI and GOME2) data. The subject is suitable for publication in AMT and the results are of interest for the users of these data. The paper is well organised and provides valuable information about TROPOMI O3 data. The analysis of error sources is interesting. It clearly highlights the limitations of the CCD retrieval method for UV sensors based on strong assumptions about the variability of the stratospheric column of O3 and on the deep convective cloud cover. I recommend this paper after the issues listed below are dealt with.

This is an excellent paper, and should be published, essentially "as is". I have a few minor suggestions, listed below, that the authors may wish to consider.

**Minor comments**

Line 62: "… systematic error and an uncertainty…"? Do you mean "systematic and random uncertainty? An error is usually something you can measure, and correct.

> This is indeed what we meant. However, we adhere to the VIM/GUM standard terminology by the Joint Committee for Guides in Metrology ([http://www.bipm.org/en/publications/guides/vim.html](http://www.bipm.org/en/publications/guides/vim.html)). See definitions 2.16, 2.18 and 2.26 on *error*, *bias* and *uncertainty*.

> We slightly reworded the original phrasing to "… target a bias and an uncertainty () …" and added the footnote : "The ESA documentation uses "bias" and "random error" but the latter is not retained here since several non-random components contribute to the uncertainty. Here, we use the VIM/GUM terms bias (estimate of a systematic error) and uncertainty (non-negative parameter that characterises the dispersion of the quantity values)."

Line 93: "Rather THEY ARE…" (Data are plural).

> Corrected as suggested.

Line 117: "…since THEY ARE not publicly released"

> Corrected as suggested.

Line 137: "of systematic error in TrOC lies in a SYSTEMATIC DIFFERENCE BE TWEEN clear-sky and fully cloudy TOC. While IT IS challenging…"

> These sentences were rephrased following the withdrawal of our statement about a cloud-dependent bias in TOC data.

Line 158: The terms "ex ante" and "ex post" will not be familiar to most readers and the authors might like to define them.

Good suggestion. Both definitions were added to the manuscript. "In the following, we use the terms ex ante uncertainty and ex post uncertainty to distinguish the reported uncertainty from the uncertainty estimated from comparisons to other observations"

Line 164: "AN effective vertical resolution…"

Corrected as suggested.

Lines 184-185: "In reality…". This sounds as though the authors feel that the homogenization process is oversold. It would be more positive to say something like "This has reduced residual systematic differences to about 5%…".

Rephrased as suggested.

Line 239: You might wish to comment on the variability you expect to see within this three-day window, or at least refer to section 4.1.3.

We added a reference to Sections 4.1.2 and 4.1.3.

Line 273: "In general, the estimation of measurement uncertainty is more intricate than the measurement itself." An odd statement, and I'm not sure how you would justify it.

See our response in the next comment.

Line 275: "It is therefore good practice not to use these uncertainty estimates blindly." A bit patronizing… unfortunately the blind don't know what they can't see.

We rephrased lines 273-275 to "Calculating accurate measurement uncertainties generally remains a challenge. Reported uncertainties (ex ante) are often first order approximations that, at times, fail to include important, poorly understood sources of error. It is therefore good practice to use ex ante uncertainty estimates with care."

Line 285: Delete "component in the".

Corrected as suggested.

Line 300: "to THE TROPOMI overpass time".

Corrected as suggested.

Lines 302-309: Can you give a rough estimate of the range of magnitude of this error? Is it important?

It is a challenge to assess the uncertainty due to differences in vertical smoothing. We reply in more detail to a similar question by Reviewer #1.

Line 319: "THE smallest random uncertainty…"

Corrected as suggested.

Line 320: Is there any other type of footprint than on the ground?

Indeed, that is redundant information. We removed "ground".

Line 352: "…cyclic pattern, OR whether it was…"

Corrected as suggested.

Line 384-387: "More likely" seems a strong statement. While it is possible, it is a speculation, and the Stauffer paper deals with stratospheric differences.

This statement was toned down to "Instead, we suspect that residual instrument-related biases exist between the ozonesonde stations".

The reference to Ryan Stauffer's findings was rephrased to "A similar grouping in the bias between ozonesonde sites has also been noted in total ozone column comparisons between sonde and satellite (OMI and Suomi-NPP OMPS-LP) (R. Stauffer, private communication, 2021)."

Lines 414-429: This seems like an important result that should be noted in the abstract.

The statement was withdrawn since we discovered that zenith-sky observations by Brewer had unintentionally slipped into the analysis. These Brewer data led to a CF-dependence which corroborated the SAOZ results. When restricting the Brewer data to direct-sun measurements only, we find no CF-dependence in S5P TOC data (see figure below). The CF-dependence in SAOZ comparisons are therefore believed to be mainly caused by a (known) cloud dependence in the SAOZ data itself.

Using the updated Brewer data, we estimate the TROPOMI TOC bias CF-dependence (i.e., the difference CF<0.1 and CF>0.8) at less than 0.2±0.2 % (1σ, standard error of the mean). This translates into at most a negative 0.4±0.4 DU (1σ, standard error of the mean) systematic error in tropospheric ozone. The relevant figure and paragraphs in the manuscript have been modified accordingly.

[Figure]

Line 448: Not sure what a "meridian dependence" is. Do you mean "latitudinal dependence"?

Yes, indeed.

Line 464: "...none of these should have superior calibration..." Is this a condition or assumption for the validity of the method?

We removed the statement as it misleads the reader into thinking it is a necessary condition, which is not the case.

Lines 222, 224 555: "between THE surface".

Corrected as suggested.

Line 55: "cloud-free".

Corrected as suggested.

Line 587: "differ"

Is this the correct line number? We couldn't find where this change is needed.

Line 586: "THE location... two-year..."

Corrected as suggested.

Line 589: "...lack of ozonesonde stations..."

Corrected as suggested.

Lines 658, 682: Semicolon, not comma.

Corrected as suggested.

Lines 670-671: This is different from the vertical integrations used by GOME and OMI, which integrate from the surface to 10 km. I didn't see any discussion of these differences. Did I miss something?

Section 3.2 states that the GOME-2B and OMI TrOC data used here are calculated using the same top level as for TROPOMI: 270 hPa. This level lies, on average, around 10.5 km in the tropics (now mentioned in Sect. 2.2 in the revised manuscript). The confusion is perhaps caused by our statement in Sect. 3.2 that the CCI tropospheric ozone data records are generated for a 10 km top level. However, we also stated there that this data version is not used for this analysis.

FYI: a quick analysis based on integrating tropical ozonesonde profiles indicates about 0.5 DU difference between the surface-270 hPa O3 column and the surface-10 km O3 column.

Figure 4: I think the caption should define Q50 and IP68. "Blue line and area show median (Q50) and 68% interpercentile (IP68) over the entire period."

Added as suggested.

Figure 5: The yellow dashed line is hard to see.

The yellow dashed line is indeed a challenge to see in print. We will improve its visibility in the revised manuscript.

Figure 14: What does "3-pentad running mean" describe exactly? I presume it's 15 of something… but it mostly tells me that the authors wish to display their knowledge of esoteric English!

This paragraph is completely rewritten after additional analysis and now avoids the use of the *pentad* terminology which is used by NOAA. It is a commonly used term in the meteorology community (https://glossary.ametsoc.org/wiki/Pentad).